# From Mixing to the Large Scale Circulation: How the Inverse Cascade Is Involved in the Formation of the Subsurface Currents in the Gulf of Guinea

Fernand Assene [1,2,*][iD], Yves Morel [1,*][iD], Audrey Delpech [1][iD], Micael Aguedjou [1,3][iD],
Julien Jouanno [1][iD], Sophie Cravatte [1][iD], Frédéric Marin [1], Claire Ménesguen [4][iD],
Alexis Chaigneau [1,3,5][iD], Isabelle Dadou [1][iD], Gael Alory [1][iD], Ryan Holmes [6][iD],
Bernard Bourlès [7][iD] and Ariane Koch-Larrouy [1,8][iD]

[1] LEGOS, Université de Toulouse, CNES, CNRS, IRD, 31400 Toulouse, France;
    audrey.delpech@legos.obs-mip.fr (A.D.); micael.aguedjou@legos.obs-mip.fr (M.A.);
    julien.jouanno@legos.obs-mip.fr (J.J.); sophie.cravatte@legos.obs-mip.fr (S.C.);
    frederic.marin@legos.obs-mip.fr (F.M.); alexis.chaigneau@ird.fr (A.C.);
    isabelle.dadou@legos.obs-mip.fr (I.D.); gael.alory@legos.obs-mip.fr (G.A.);
    ariane.koch-larrouy@legos.obs-mip.fr (A.K.-L.)
[2] Department of Oceanography, Institute of Fisheries and Aquatic Sciences, University of Douala at Yabassi,
    Douala-Bassa BP 7236, Cameroon
[3] International Chair in Mathematical Physics and Applications (ICMPA-UNESCO Chair), University of
    Abomey-Calavi, Cotonou 072BP50, Benin
[4] LOPS, Université de Bretagne Occidentale, Ifremer, CNRS, IRD, 29280 Plouzané, France;
    claire.menesguen@ifremer.fr
[5] Institut de Recherches Halieutiques et Océanologiques du Bénin (IRHOB), Cotonou 03BP1665, Benin
[6] CCRC, ARC CLEX and the School of Mathematics and Statistics, University of New South Wales,
    Sydney NSW 2052, Australia; ryan.holmes@unsw.edu.au
[7] IMAGO, IRD, 29280 Plouzané, France; bernard.bourles@ird.fr
[8] Mercator Ocean, 31520 Ramonville-Saint-Agne, France
[*] Correspondence: fernandrosvelt@gmail.com (F.A.); yves.morel@legos.obs-mip.fr (Y.M.);
    Tel.: +33-561-333-055 (Y.M.)

**Abstract:** In this paper, we analyse the results from a numerical model at high resolution. We focus on the formation and maintenance of subsurface equatorial currents in the Gulf of Guinea and we base our analysis on the evolution of potential vorticity (PV). We highlight the link between submesoscale processes (involving mixing, friction and filamentation), mesoscale vortices and the mean currents in the area. In the simulation, eastward currents, the South and North Equatorial Undercurrents (SEUC and NEUC respectively) and the Guinea Undercurrent (GUC), are shown to be linked to the westward currents located equatorward. We show that east of 20° W, both westward and eastward currents are associated with the spreading of PV tongues by mesoscale vortices. The Equatorial Undercurrent (EUC) brings salty waters into the Gulf of Guinea. Mixing diffuses the salty anomaly downward. Meridional advection, mixing and friction are involved in the formation of fluid parcels with PV anomalies in the lower part and below the pycnocline, north and south of the EUC, in the Gulf of Guinea. These parcels gradually merge and vertically align, forming nonlinear anticyclonic vortices that propagate westward, spreading and horizontally mixing their PV content by stirring filamentation and diffusion, up to 20° W. When averaged over time, this creates regions of nearly homogeneous PV within zonal bands between 1.5° and 5° S or N. This mean PV field is associated with westward and eastward zonal jets flanking the EUC with the homogeneous PV tongues corresponding to the westward currents, and the strong PV gradient regions at their edges corresponding to the eastward currents. Mesoscale vortices strongly modulate the mean fields explaining the high spatial and temporal variability of the jets.

**Keywords:** equatorial atlantic; NEUC (North Equatorial Undercurrents); SEUC (South Equatorial Undercurrents); vortices; mixing; friction

## 1. Introduction

The circulation in the upper layers of the Tropical and Equatorial Atlantic forms a complex system of zonal currents, horizontally and vertically stacked (see for instance [1] for a detailed description). It has a strong influence on the AMOC (Atlantic meridional overturning circulation), West African monsoon and, in the eastern Atlantic margin, the subsurface oxygen minimum zone [2–7]. Several observational international programs have been dedicated to the study of the tropical Atlantic. In particular, the Prediction and Research Moored Array in the Tropical Atlantic (PIRATA, see [8,9]) mooring system has been deployed in the framework of an international cooperation since 1993. Annual oceanographic cruises are also conducted within this project. They allowed a better description of the spatial and temporal variability of the surface and subsurface currents [10–14]. Other programs have recently been carried out [15,16], or are underway (see the South and Tropical Atlantic climate-based marine ecosystem prediction for sustainable management, "TRIATLAS" project, https://triatlas.w.uib.no/).

Despite these recent efforts, there is still a lack of observations in this region, in particular at the mesoscale. For instance, the dynamics of the intrathermocline current system is still debated. Studies have mostly focused on the eastward subsurface currents, the equatorial undercurrent (EUC) and the NEUC and SEUC (sometimes called the Atlantic Tsuchiya current for comparison with their Pacific counterparts, see [17]). Numerical models are able to reproduce realistic current systems and have been used to study the origin and fate of the latter currents [2]. Hazeleger et al. [3,4] showed that most of the water parcels entering the EUC come from the Southern Hemisphere, with only a fraction coming from the Northern Equatorial current (NEC). This asymmetry is due to the influence of the Atlantic Meridional Overturning Circulation [18]. EUC waters come principally from the Southern Hemisphere, flowing within the North Brazil Current along the Brazilian coast before veering eastward, forming the EUC [1,19]. They are then upwelled during their journey along the Equator from the western to the eastern Atlantic. They recirculate within the upper equatorial currents system before leaving the tropical area and entering the general overturning circulation. Wang [20] explained the formation of the SEUC and NEUC by adiabatic meridional advection of particles with PV anomalies recirculating from the EUC. Huttl-Kabus et al. [21] showed that the cores of NEUC and SEUC mostly originate from the southeastern tropical Atlantic. Jochum and Malanotte-Rizzoli [22] used idealized simulations to show that tropical instability waves (TIW) helped to maintain the SEUC against dissipation west of 10° W. Greatbatch et al. [23] showed that intraseasonal variability ($\simeq$10 s of days) provides the energy to maintain semiannual and longer time scale ocean currents against dissipation, but they focused on the Equatorial Deep Jets (EDJ). In the Gulf of Guinea, there is also a strong seasonal cycle associated with the annual equatorial upwelling forced by the trade winds and the formation of the so called cold-tongue from June to August and a second shorter season from November to December. During this event, the complex system of zonal currents is modulated in particular with a weaker EUC [14,24].

Observations have also revealed the existence of westward subsurface currents between the EUC and the SEUC and NEUC [11,25]. In the Eastern basin, the northern one is also named Guinea countercurrent and their signatures are intensified from December to June. Their origin has received less attention and they are generally assimilated to a deep signature of the South Equatorial Current (SEC) on both sides of the EUC (forming two branches called the sSEC and nSEC) or they are associated with recirculation branches connecting the EUC with the SEUC and NEUC [1,24]. The vertical and geographical position of all the currents discussed in the present paper are indicated in Figures 1 and 2.

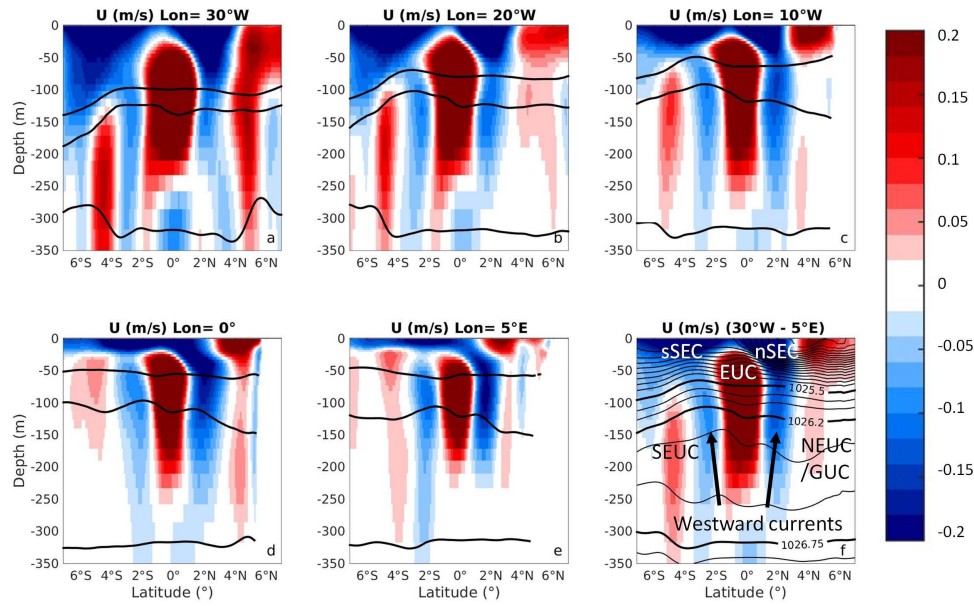

**Figure 1.** Meridional sections of the zonal velocity *U* averaged over 2015 at longitudes 30° W, 20° W, 10° W, 0°, and 5° E (**a**–**e**) and zonal mean (30° W to 5° E) of the temporal averaged (**f**). Thick black lines are associated with the position of isopycnic surfaces with potential density $\rho = 1025.5$, 1026.2 and 1026.75 kg · m$^{-3}$ defining layers 1 and 2. Thin black lines on the zonal mean are additional density contours (step $\Delta\rho = 0.2$ kg · m$^{-3}$).

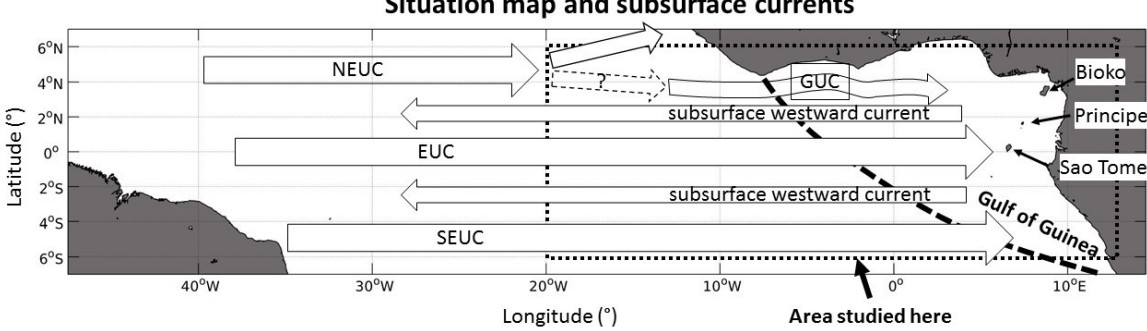

**Figure 2.** Situation map of the studied area. The position and name of the main subsurface currents discussed in the paper are indicated. The dotted box represents the area on which we focus.

In this paper, we analyse the results from a numerical model of the tropical Atlantic at 1/12° horizontal resolution, described in Section 2. We focus on the formation and dynamics of the westward and eastward (SEUC and NEUC) subsurface currents, on both sides of the Equator in the Gulf of Guinea. We first show that the characteristics of the subsurface currents are coherent with observations from previous studies (Section 3.1). We then concentrate our analysis on two layers where the SEUC, NEUC/GUC and westward currents are intensified. We show that, in the model, the mean currents are associated with homogeneous regions of mean potential vorticity (PV). Mesoscale anticyclonic vortices are involved in the formation and maintenance of the homogeneous PV tongues and explain the strong spatial and temporal variability of the zonal currents (Section 3.3). We then focus on the formation of the vortices, taking place in the eastern part of the Gulf of Guinea. Qualitative analysis show that meridional advection and diabatic effects, such as mixing and friction, can be all involved in the formation of the PV anomalies of fluid parcels forming the vortex cores (Section 3.5). Our results are summarized and their limits are discussed in the concluding section.

## 2. Model, Configuration and Diagnostics

### 2.1. The Tropical Atlantic NEMO Configuration

The numerical simulation we analyse is based on an eddy-resolving configuration of the equatorial and Southern Tropical Atlantic (32° S to 15° N, 60° W to 20° E) at 1/12° horizontal resolution, embedded in a coarser eddy-permitting Tropical Atlantic configuration (35° S to 35° N, 100° W to 20° E) at 1/4° horizontal resolution. Tides are not taken into account. Even though internal tides can induce important mixing close to continental margins, a resolution of 1/12° remains too coarse to correctly represent them. We use the General Bathymetric Chart of the Oceans (GEBCO) bathymetry [26] interpolated on each grids. The numerical code is the oceanic component of the Nucleus for European Modeling of the Ocean program (NEMO, [27]). It solves the primitive equations discretized on a C-grid and fixed vertical levels (z coordinate). The bottom topography is thus represented by staircases and there exist lateral walls, with possible development of lateral viscous layers at all depths (see Section 3.4.3). The two domains are coupled online via the AGRIF library in two way mode [28,29]. They share the same vertical grid which consists of 75 levels in the vertical, with 12 levels in the upper 100 m and cell thickness reaching 175 m near the bottom. The thickness of the bottom cells is adjusted to improve the representation of the bottom topography. A third-order upstream biased scheme (UP3, [30]) is used for momentum advection, with no explicit horizontal diffusion. A Total Variance Dissipation scheme (TVD) is used for tracers together with a Laplacian isoneutral diffusion of $300 \, \mathrm{m^2 \, s^{-1}}$ for the coarse grid and $125 \, \mathrm{m^2 \, s^{-1}}$ for the fine grid. The temporal integration is achieved by a modified leap-frog Asselin filter [31], with a coefficient of 0.1 and a time step of 1350 s for the coarse grid and 450 s for the fine grid.

The vertical diffusion coefficients are given by a turbulence closure scheme (TKE, [32]). Bottom friction is quadratic with a bottom drag coefficient of $10^{-3}$ and free slip boundary conditions are applied at the lateral boundaries. At the lateral open boundaries of the coarse grid, the temperature, salinity, horizontal velocities and sea level are forced using an interannual hindcast from the MERCATOR global daily reanalysis GLORYS2V4 [33]. At the surface, the atmospheric fluxes of momentum, heat, and freshwater are provided by bulk formulae [34] and forced with the DFS5.2 product [35], which is a bias corrected version of ERA-Interim reanalysis [36]. It consists of 3 h fields of wind speed, atmospheric temperature and humidity, and daily fields of long wave, short wave radiation and precipitation. The shortwave radiation forcing is modulated online by an analytical diurnal cycle. A monthly climatological runoff based on the data set of [37] is prescribed near the river mouths. Note that there is no restoring of the model toward observed or climatological salinity and temperature. The simulation was initialized with temperature and salinity from GLORYS2V4 on 1 January 1993 and performed over the period 1993–2015. In this study, we restrict our attention to year 2015 and focus on mechanistic aspects of the dynamics given by the model and on the seasonal variability, which is generally stronger than the interannual variability [38,39]. We selected 2015 as, after 22 years, an equilibrium has been reached for the seasonal cycle (we verified that after about three years of simulations the kinetic energy is stabilized) and 2015 is a neutral year compared to the interannual variability in the equatorial Atlantic ocean (see [39] and https://www.ospo.noaa.gov/Products/ocean/sst/anomaly/2015.html). The regional coarse grid configuration has proven its ability to properly resolve equatorial dynamics and associated seasonal and interannual variability (e.g., [38,40,41]). We expect the high-resolution grid to perform equally or better, especially in resolving the sheared equatorial current system.

### 2.2. Potential Vorticity Diagnostics

For geophysical flows, isopycnic variations of PV determine the currents [42]. For instance, it has been shown that the jet-like structure of equatorial currents are associated with zonal bands of alternatively low and high PV gradients [43–45], with low gradients associated with westward

currents and high gradients to eastward currents. Since PV is conserved for adiabatic dynamics [46,47], the guideline of our study is to explain how the latter PV structure is formed and maintained.

We thus focus on the evolution of the PV field, but we first follow [48] (see also [45]) and define a rescaled PV

$$
\begin{aligned}
PV_{rescaled} &= -(\vec{\nabla} \times \vec{U} + \vec{f}).\vec{\nabla} Z(\rho), \\
&= -div(\,(\vec{\nabla} \times \vec{U} + \vec{f})\, Z(\rho)\,)
\end{aligned}
\tag{1}
$$

where $\vec{U}$ is the velocity field, $\vec{f}$ is the Earth rotation vector, whose projection on the local vertical axis defines the Coriolis parameter $f$, and $Z(\rho)$ is a function of the potential density. As already mentioned by [46] (see also [47]), $PV_{rescaled}$ has the same properties as the traditional Ertel PV, obtained with $Z(\rho) = \rho$. The problem for the traditional form is that vertical sections are dominated by the signature of the pycnocline, and the dynamical signal associated with isopycnic variations of PV is difficult to identify. To overcome this, we chose $Z(\rho^*) = z$ for a specific location, where the density profile $\rho^*(z)$ is typical of the stratification of the area and can be taken as a reference to rescale the PV.

The rescaled PV is in fact close to the quasigeostrophic PV. It scales as a vorticity with a reference value at rest close to $f$ (see [45,48]). Deviations of the rescaled PV from its background value at each latitude (also referred to as PV anomalies) are the signature of vortical geostrophic circulation patterns [45,48] and is easier to interpret. For instance, along isopycnic surfaces, local maxima or minima of the rescaled PV are associated with vortices (see [49–52]), a key property that can be used to detect and separate them from the vorticity signature of gravity waves such as equatorial Kelvin waves, which have a strong signature in the equatorial Atlantic ocean [53]. The dynamical signature of a PV anomaly is also non-local: A PV anomaly located within some density range is associated with a vorticity and currents that extend over the whole water column but with the strongest signature within the layer of the PV anomaly and decreasing away from it. The consequences are that vorticity or currents in a layer can be associated with PV anomalies from different layers, but the existence of a vorticity or current maximum within a layer is necessarily associated with a PV anomaly in the same layer [51,52].

In our case, the chosen profile is located at [8.3° E, −1° N] on 10 May, as the location and time correspond to a situation for which the surface density reaches low values, which is adequate to rescale PV in the layers we study. Hereafter, PV refers to the rescaled PV.

## 3. Results

### 3.1. Mean Zonal Subsurface Currents in the Equatorial Band

Here, we concentrate our attention on the equatorial band, between 6° S and 6° N, and mostly on longitudes between 20° W and 10° E or in the Gulf of Guinea (see Figure 2). Analysis of vertical sections of PV show that, in this area, anomalies mostly appear in the potential density range [1025.5, 1026.75] kg·m$^{-3}$. This corresponds to depths extending from the lower part of the pycnocline to the less stratified waters just below ($\simeq$50 to 250/300 m). Because the currents, vorticity, and PV anomalies exhibit vertical variations within this range, we split it into two layers. Layer 1, with the density range [1025.5, 1026.2] kg·m$^{-3}$, corresponds to the lower part of the pycnocline. Layer 2, with the density range [1026.2, 1026.75] kg·m$^{-3}$, is localized just below the pycnocline and is thus less affected by the surface dynamics.

Figure 1 shows meridional sections of the time averaged zonal velocity for 2015 at different longitudes as well as its longitudinal mean. The EUC has a clear signature at all longitudes and spreads across the pycnocline and below. The colorbar has been chosen to focus on the zonal currents flanking the EUC, hereafter called the lateral currents, and is thus saturated for the EUC, but its order of magnitude is coherent with observations (10–15 Sv, decreasing eastward, see [54,55]).

The lateral subsurface currents exhibit a zonal jet structure with clear signals of the SEUC and NEUC around 5° S and 5° N. In the northern part, east of 10° W, we find the GUC, which is sometimes presented as disconnected from the NEUC [11,24]. Eastward jets are systematically accompanied

by westward counterparts flanked equatorward. Whereas these westward currents are generally considered as deep extensions of the broad westward SEC, cut in two branches by the eastward EUC, it appears here that the westward currents are of the same magnitude as the SEUC, NEUC or GUC, and have subsurface cores. Eastward and westward lateral currents have their cores located within the [1025.5, 1026.75] kg · m$^{-3}$ density range. The meridional and vertical positions of the jet cores are variable zonally as well as their strength, ranging from 5 cm/s to 15 cm/s. In particular, the vertical position of the cores of the currents seems to alternate between layers 1 and 2 (thick black lines).

Figure 3 represents the zonal velocity at 10° W on the 15th of each month. The different lateral subsurface currents are usually well observed but their strength varies from 0 (meaning the zonal current vanishes) to ±25 cm/s. Note in particular their weakening between June and August, when the equatorial upwelling intensifies (note the ascent of the pycnocline) accompanied by the weakening of the EUC weakens and the intensification of the westward SEC [55]. Again, the vertical position of the cores of the currents alternates between layers 1 and 2 (thick black lines).

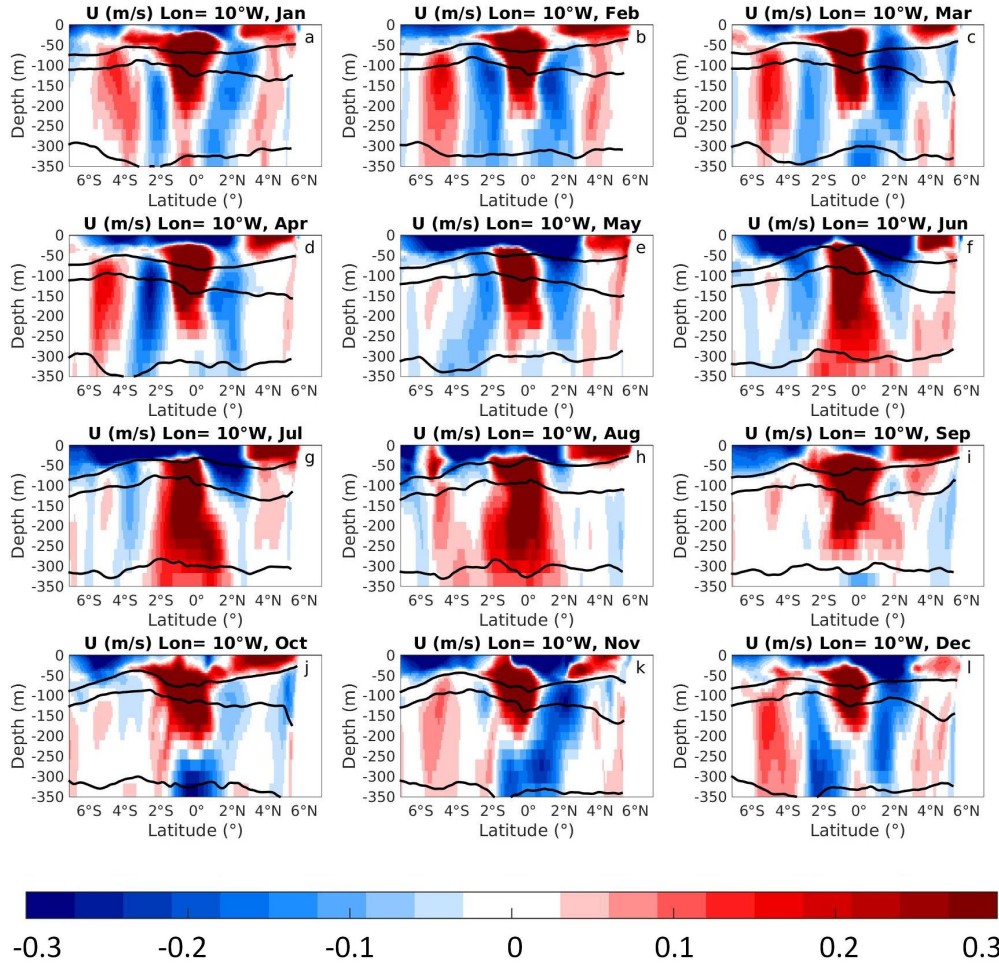

**Figure 3.** Meridional sections of zonal velocity *U* at 10° W on the 15th of each month (instantaneous fields) for 2015. The colorbar ranges from −0.3 to 0.3 m/s. Thick black lines are associated with the position of isopycnic surfaces with potential density $\rho$ = 1025.5, 1026.2 and 1026.75 kg · m$^{-3}$.

Strong variability of instantaneous currents (strength, meridional and vertical positions) are characteristic of observations in the region [11–14], but previous studies focused on eastward currents. In particular, the variability for the NEUC, is in agreement with [56] who observe a weak seasonal cycle of NEUC transport at (5° N, 23° W) but strong intra-seasonal events. Rosell-Fieschi et al. [57]

(see also [58,59]) observe a nSEC intensification in summer and early winter, and a decrease (which can even lead to reversal) in spring due to surfacing of the EUC. North of the nSEC, the NECC is subject to very strong seasonal variability too, with the strongest eastward speeds between April and September. This seasonal variability of surface currents is forced by the seasonal variations of winds, in connection with the meridional shift of the Intertropical Convergence Zone (ITCZ). The high seasonal variability of the EUC is associated with the equatorial upwelling season (May to August in the present simulation). It leads to the vanishing of the upper part of the EUC (above 1025.5 kg/m$^3$) and to a deepening of its signature. The high variability at the equator below 200 m has also been associated with the seasonal variability of the Equatorial Intermediate Current (EIC) due to Rossby waves [60]. All these behaviors are reproduced by the simulation. However, Kolodziejczyk et al. [13] observe strong variability of the SEUC transport and maximum velocity but no clear seasonal cycle. While strong variability is observed in the simulation too, the maximum current of the EUC is much stronger in the observations (55 cm/s) and the absence of seasonal cycle is in contradiction with the present results which exhibit a strong weakening during the upwelling season. This discrepancy could be due to a limited number of cruises (17) in [13], which may be too low to describe a seasonal cycle.

To conclude, the mean and instantaneous velocity sections presented in Figures 1 and 3 are reasonably close to existing observations. In particular, both the model and the observations show that eastward and westward subsurface currents reach their maximum within layers 1 or 2, indicating that an important part of their signal is associated with PV anomalies located within these layers, on which we thus focus in the following.

## 3.2. Mean Zonal Currents and PV Fields

Figure 4 represents the mean zonal velocity in layers 1 and 2 (in color). The zonal velocity field has been averaged between isopycnic levels determining the layer, and over time for the whole year 2015. Note the jet-like structures of the currents. In the upper layer 1, the westward currents are intensified in the Gulf of Guinea, decreasing westward, but some local intensification are observed in the TIW area, between 10 and 30° W, and close to the western boundary for the northern branch. The SEUC reaches its maximum between 10° W and 0°, but vanishes in the western part of the basin. The NEUC is strong at west and becomes much weaker east of 20° W, as it deviates poleward. It does not enter in the Gulf of Guinea, but the GUC is found East of 10° W, in agreement with [11]. Below, in layer 2, the mean structure of the currents is similar but less intense. The main difference is the SEUC intensification at West. Globally, there is a clear difference in the dynamics east and west of 20° W, which is the western limit of the cold-tongue associated with the equatorial upwelling [61,62]. In the following, we will concentrate on the Gulf of Guinea (east of 20° W). The dynamics in the western area are more complex, involving other layers, and will be discussed in the final section.

The isolines of the mean PV field (averaged between isopycnic levels and over the whole year) have been superimposed on the mean zonal currents to underline their tight relationship: eastward jets are associated with strong PV gradients, while westward jets are associated with homogenized PV fields. In fact, in the open ocean, when (cyclo)geostrophic equilibrium is the main dynamical constraint, there is a tight connection between isopycnic PV variations (or anomalies with respect to PV at rest) and currents (see [42,48,63,64] and references therein). This holds even close to the Equator in the latitude band we consider [43,45]. As already mentioned, this relationship is non-local, currents depend on the PV anomalies over the entire water column, but decreasing away from the area of PV variations. In the present case, the strong correlation between PV variations and the zonal currents in each layer shows that the currents are mostly related to the spatial distribution of PV within layers 1 and 2.

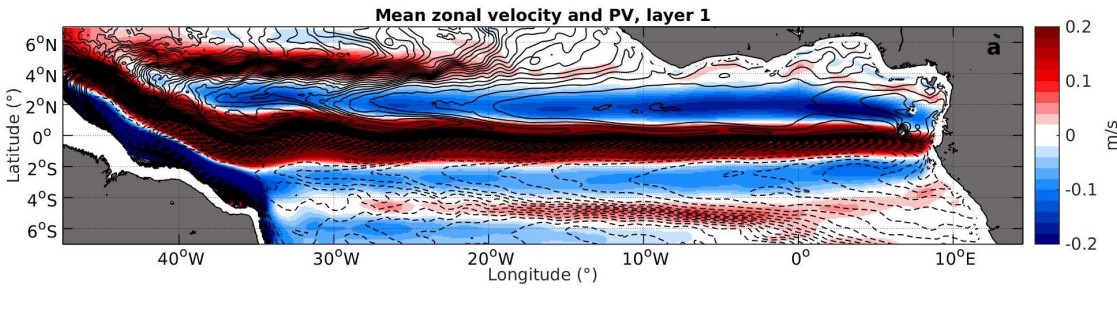

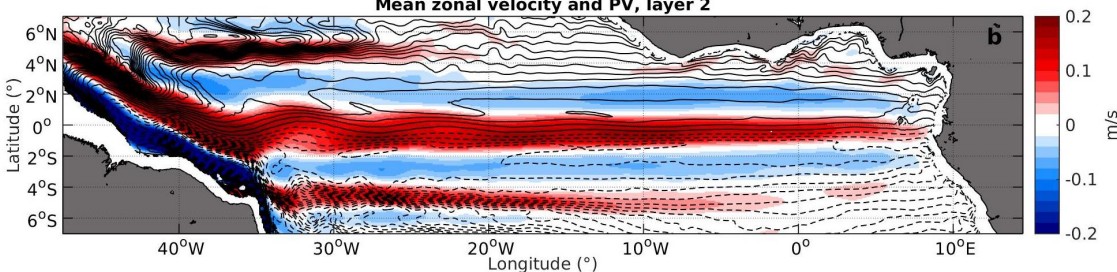

**Figure 4.** Time averaged zonal velocity $U$, vertically averaged within layer 1 (**a**), in color) and 2 ((**b**), in color). The thin black lines are isolines of the mean PV field in each layer (negative values are dashed. The contour step is $2 \times 10^{-6}$ s$^{-1}$ for layer 1 and $1 \times 10^{-6}$ s$^{-1}$ for layer 2).

The spatial distribution of PV is represented in color in Figure 5 for each layer (a and b). The latitudinal profiles of PV over the longitude band [20° W, 0° E] are also represented in Figure 6 for each layer (the thick black line represents the average profile over this longitude band). The striking feature is that in both layers and both Hemispheres, the regions of homogenized PV are associated with PV tongues originating from the eastern area of the Gulf of Guinea and extending at least up to 20° W. The current structures can thus be understood from the generation and spreading of these PV tongues.

The general salinity structure of the model (Figure 5b,d) is coherent with observations. In the western region, the SEUC and NEUC are associated with salty waters originating from the subtropics. The westward currents get their high salinity from recirculation of the EUC in the Gulf of Guinea [13,14,24]. In the upper layer, we can see that the EUC brings high salinity along the Equator, but the isopycnic salinity value decreases eastward because of diapycnal mixing, as suggested by an opposite evolution in layer 2. Mixing due to shear stress between the EUC and surface currents is known to increase surface salinity in the Gulf of Guinea [10,41,65–67]. Figure 5d shows that salinity increases along the Equator in layer 2, indicating that mixing also injects saline EUC waters into this deeper and fresher layer. A striking feature is that PV tongues are co-localized with salinity tongues originating from the Gulf of Guinea. In particular, the maximum salinity is observed at the eastern boundary of the Gulf of Guinea, corresponding to the regions of formation of the PV tongues.

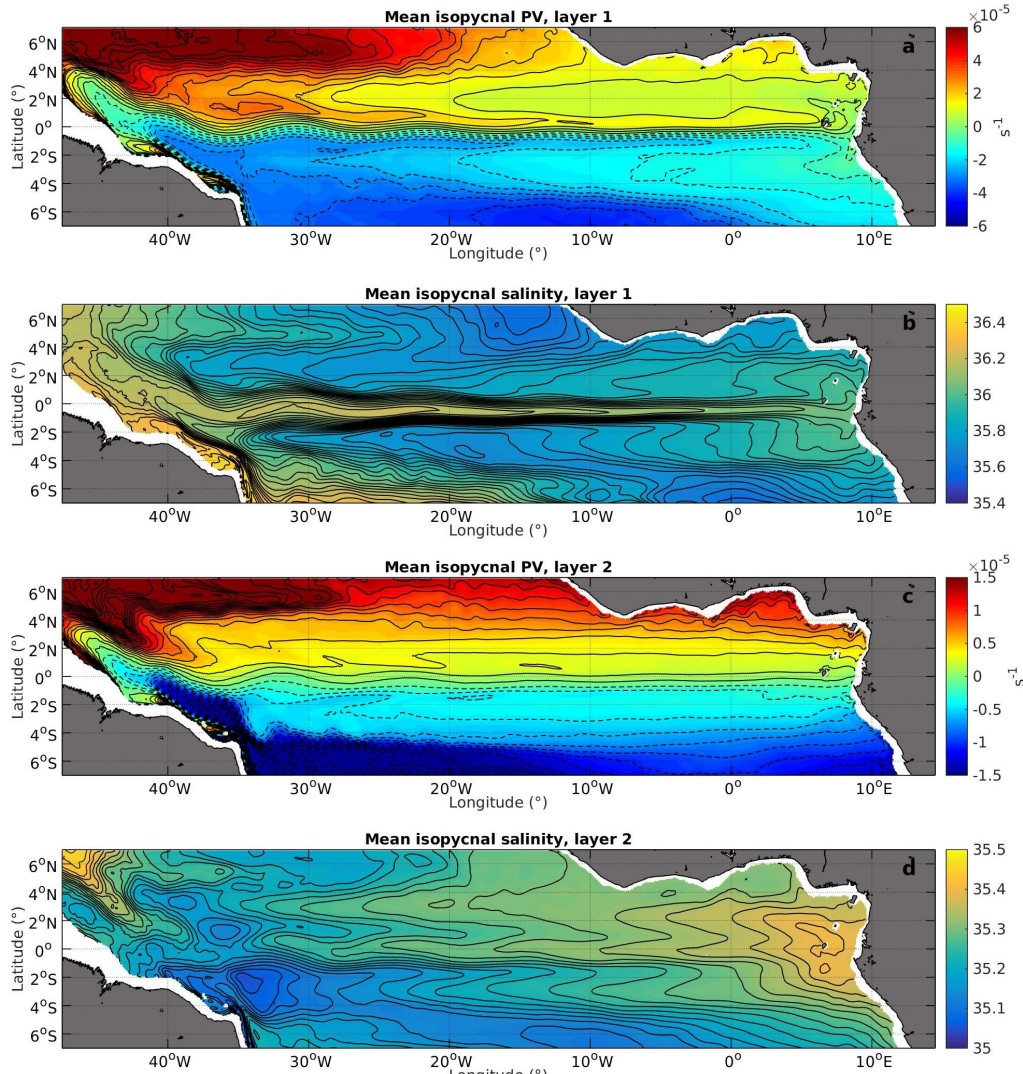

**Figure 5.** Horizontal map of the PV fields averaged over 2015 and between isopycnic levels defining layers 1 and 2 (**a**,**c**, respectively), and horizontal map of the PV fields averaged over 2015 and between isopycnic levels defining layers 1 and 2 (**b**,**d**, respectively).

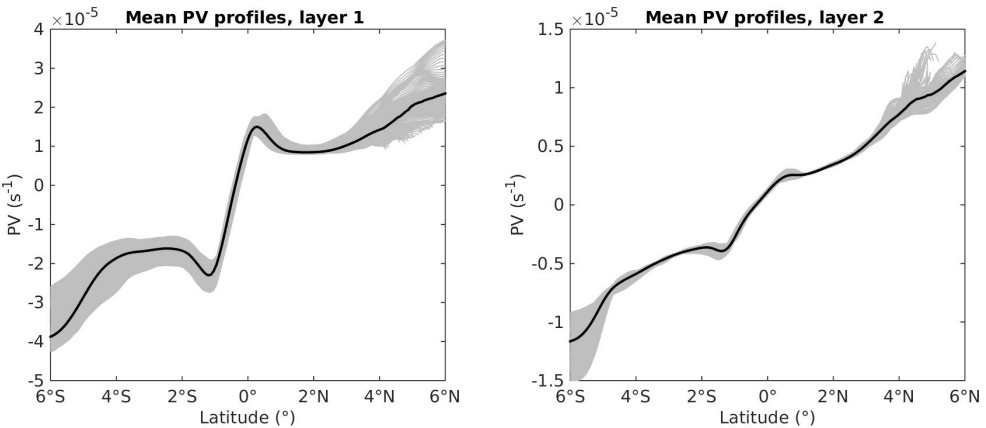

**Figure 6.** Latitudinal PV profiles over the longitude band [20° W, 0° E] for each layer (in gray). The average profile over this longitude range is the thick black line.

### 3.3. High Frequency Mesoscale Circulation

The mean fields described above yield a picture where the EUC transports salty waters into the Gulf of Guinea, which gradually mix with the deeper and fresher waters of layer 2. A high salinity anomaly appears in layer 2 and is spread zonally on both side of the equator. The mean currents are consistent with PV tongues originating from the same area as the salinity tongues.

The formation of the PV structure and the mechanisms responsible for the westward spreading of the PV tongue are fundamental to understand the formation of the zonal currents.

### 3.3.1. Analysis of Vortical Structures

Figure 7 is similar to Figure 5 but describes the instantaneous PV and salinity fields on 10 May 2015. The large scale structure is similar to the mean fields depicted in Figure 5, but the spatial variability is marked by the presence of mesoscale variability with roughly circular horizontal shapes associated with PV and salinity extrema. The PV extrema correspond to positive anomalies (with respect to the local PV value) south of the Equator and negative anomalies north of the Equator, indicating anticyclonic mesoscale structures.

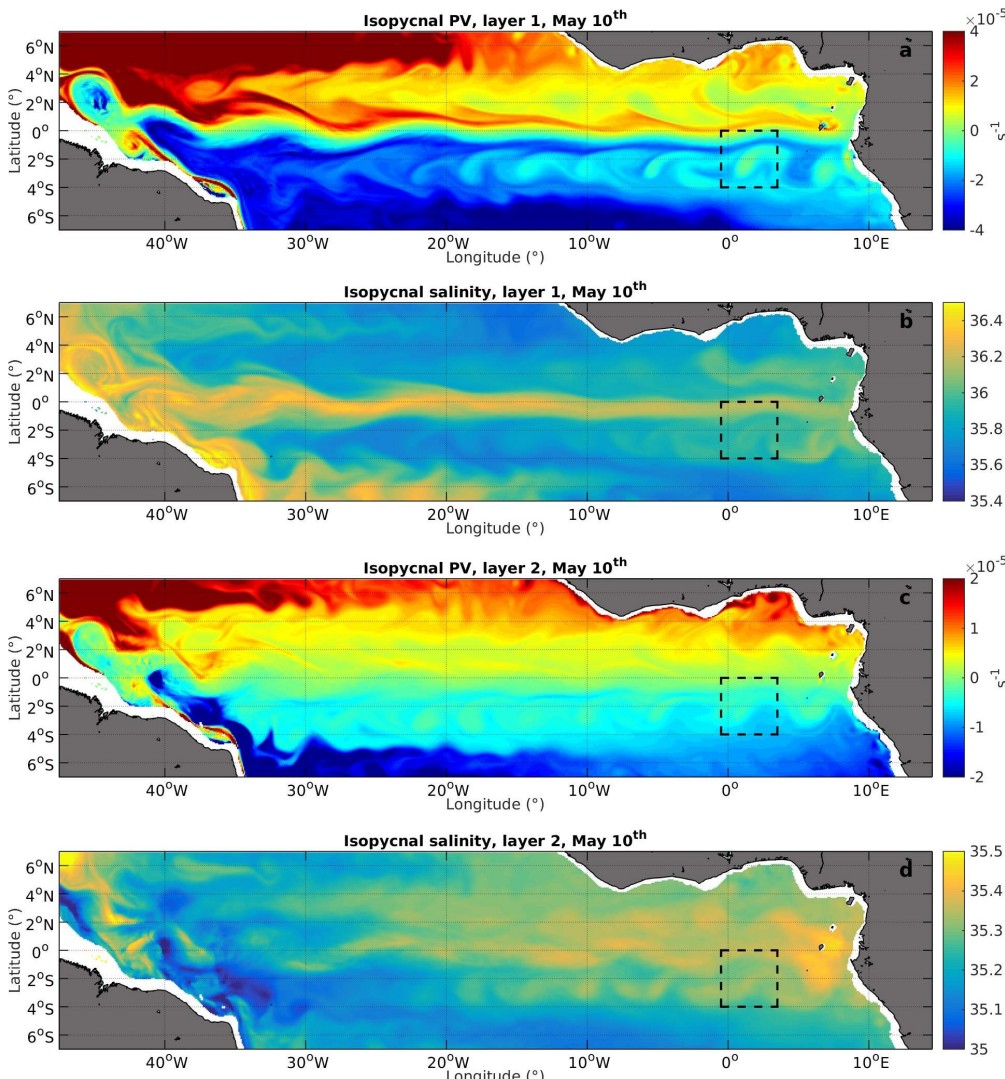

**Figure 7.** Horizontal map of the PV fields in layers 1 and 2 (**a,c**) on 10 May, and horizontal map of the salinity fields averaged between isopycnic levels defining layers 1 and 2 (panels (**b,d**)) on 10 May. The dashed box indicates the eddy that is analysed.

Figure 8 represents a Hovmöller diagram of PV along 3° S and for the whole year 2015 in layers 1 (left panel) and 2 (right panel). Anticyclonic vortices are formed in the eastern part of the equatorial band, then propagate westward for several thousands of kilometers at an average speed of about 15 cm/s, similar to all vortices. Their intensities in layers 1 and 2, and their vertical structures, are variable. All vortices are rapidly dissipated and their formation is interrupted during the upwelling season from about day 180 (early June) to 270 (late August). Similar diagrams for salinity fields (not shown) or for the Northern Hemisphere exhibit the same patterns.

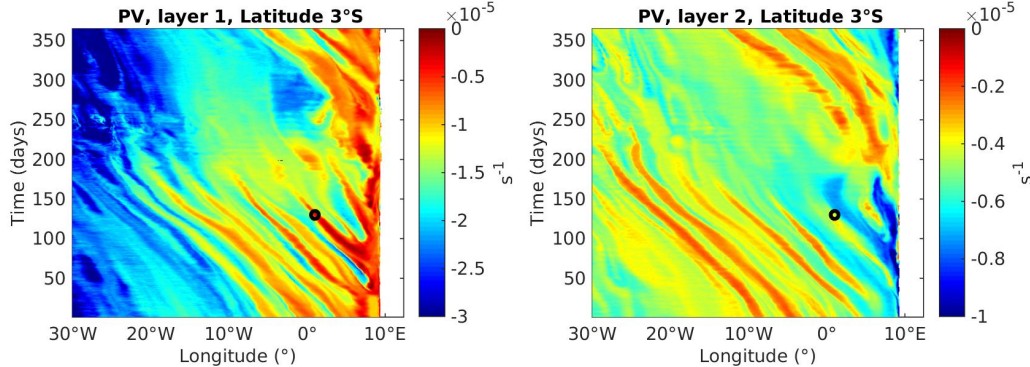

**Figure 8.** Hovmöller diagram of PV along 3° S for the whole year 2015 in layers 1 (left panel) and 2 (right panel). The black circles indicate the position of the vortex analysed in details and which is formed just before the equatorial upwelling season.

We analyse a particular structure located at [1.1° E, 2.6° S] on 10 May (see dashed box in Figure 7). As shown in Figure 8, this vortex has just been formed and is about to be dissipated. Figure 9 represents horizontal maps of its associated PV, vorticity and salinity in each layer. The maximum vorticity reaches $5 \times 10^{-6}$ s$^{-1}$ in layer 1 and $2.5 \times 10^{-6}$ s$^{-1}$ in layer 2. It is positive, corresponding to an anticyclonic vortex whose radius is about 75 km. This particular vortex, as well as most of the identified structures with closed PV contours, has a clear signature in salinity. These PV and salinity cores propagate westward over long distances (see below) and correspond to nonlinear subsurface vortices, not to Rossby waves. In addition, the PV field is not symmetric with respect to the Equator and, although vortices propagate on both sides of the Equator, they are not zonally aligned or of the same strength. This confirms that these PV anomalies correspond to nonlinear structures, able to transport the water properties in their cores, and not to equatorial Rossby waves.

Figure 10 represents the vertical structure of several properties (PV, salinity anomaly, vorticity, zonal currents) across the particular vortex shown in Figure 9 (its center is located at (1.1° E, 2.6° S)). PV is composed of two vertically aligned cores, with one maximum located in layer 1 (at a depth $\simeq$ 120 m) and the second one in layer two (at a depth $\simeq$ 200 m). PV anomalies with respect to the background reach about $10^{-5}$ s$^{-1}$ in layer 1 and $0.5 \times 10^{-5}$ s$^{-1}$ in layer 2. As expected from Figure 9, the vorticity and zonal velocity signatures are intensified in the first layer, but extend deeper than the PV anomalies. The salinity anomaly of the vortex core is positive and reaches 0.2 in layer 1 and 0.1 in layer 2 (see also Figure 9).

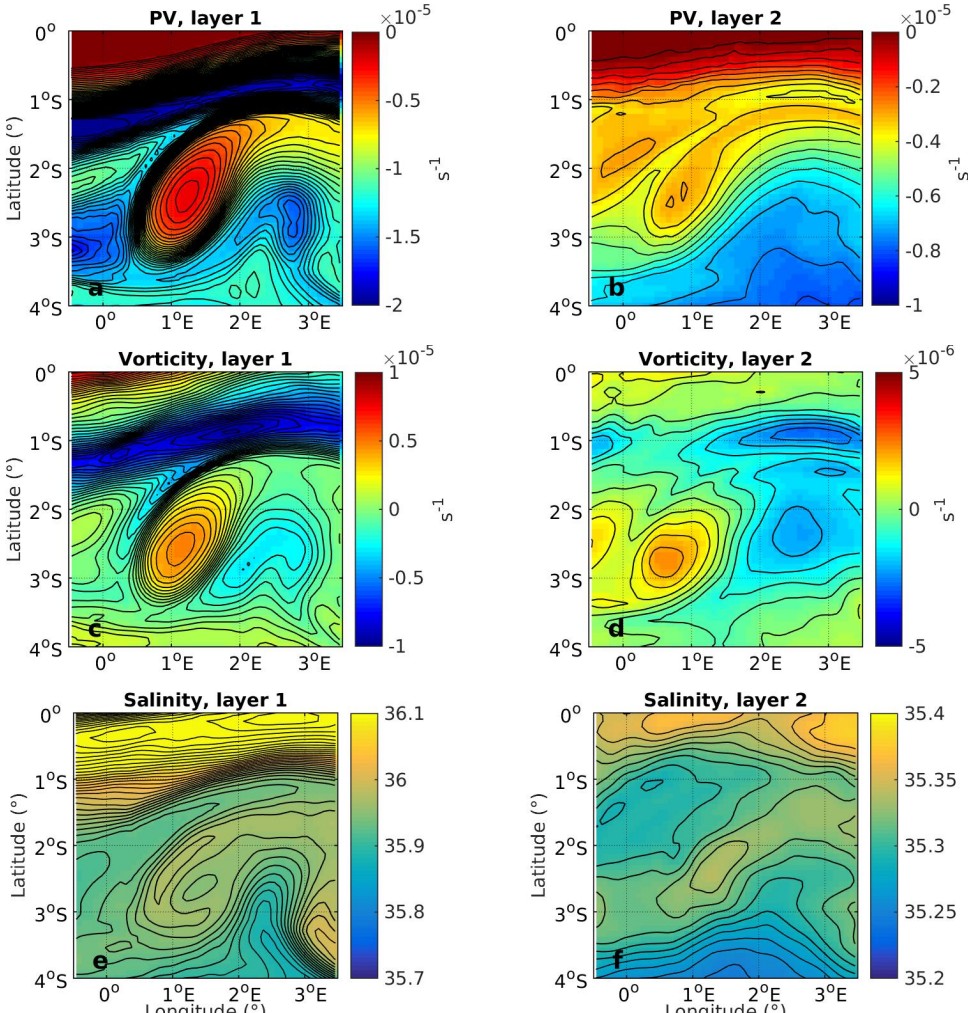

**Figure 9.** Horizontal maps of dynamical properties of a particular eddy, centered at [1.1° E, 2.6° S] on 10 May: isopycnic PV in layers 1 and 2 (**a**,**b**); isopycnic vorticity in layers 1 and 2 (**c**,**d**); isopycnic salinity in layers 1 and 2 (**e**,**f**). Steps between salinity contours is 0.01 and $5 \times 10^{-7}$ s$^{-1}$ for PV and vorticity

The zonal velocity section presented on Figure 10 for 10 May bears similarities with the section at 10° W presented in Figure 3 for 15 May, with interesting differences: the strength of the westward and eastward (SEUC) currents are close, but their vertical positions and extents are different, showing the influence of the anticyclonic eddy to the reinforcement of both currents in the upper layer. Finally, the subsurface anticyclonic vortices have no detectable signatures on sea surface height, temperature or salinity (not shown). In fact, the present simulation is consistent with satellite observations where vortices are rarely detected in the Gulf of Guinea and along the equatorial band [68]. It is only close to the coast that surface vortices have been identified, but this is not discussed further here.

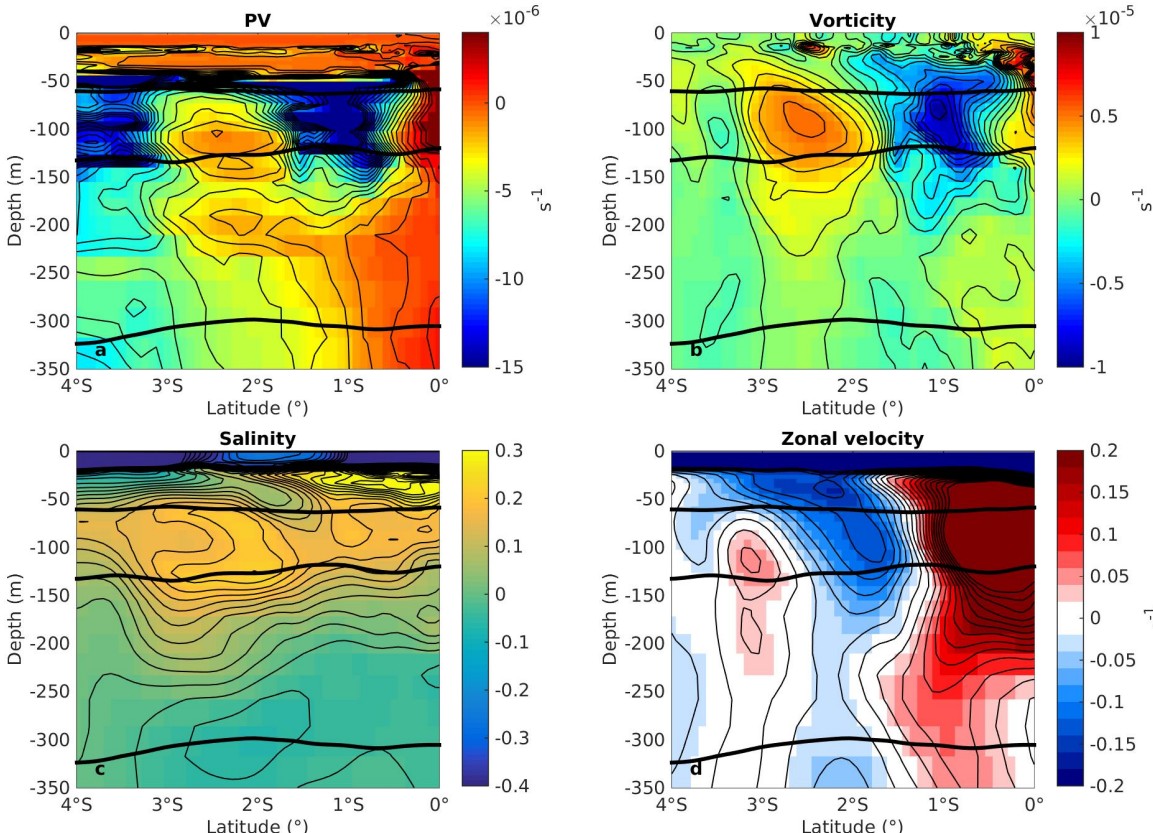

**Figure 10.** Meridional sections of PV (**a**), vorticity (**b**), salinity anomaly (with respect to the annual mean, (**c**) and zonal velocity $U$ (**d**) along 1.1° E, on 10 May. The section passes across the eddy described in Figure 9. Thick black lines represents the isopycnic levels defining layers 1 and 2. Thin lines represents isovalues (Steps: $10^{-6}s^{-1}$ for PV and vorticity, 0.025 for salinity and 2.5 cm/s for velocity).

In general, the subsurface eddies located both South ($\simeq$2–3° S) and North ($\simeq$2–3° N) of the equator are anticyclonic and share similar properties, except they have opposite signs of PV anomalies and vorticity. This can be seen in Figure 11, which represents zonal sections of vorticity and salinity along latitudes passing through anticyclonic vortices of the southern (3° S) and northern (2° N) westward currents. Note the correspondence between the vorticity cores and salinity anomalies of vortices identified as PV or salinity extrema in Figure 7. This zonal section shows that the cores of vortices are always located in the subsurface layers, but their exact vertical position varies (located in layers 1, 2 or in between), which has strong consequences on the spatial variability of the vertical structure of westward and eastward currents, as observed in situ [14,24]. Vertical sections of PV (not shown) exhibit the same structure, with main anomalies located in layers 1 or 2.

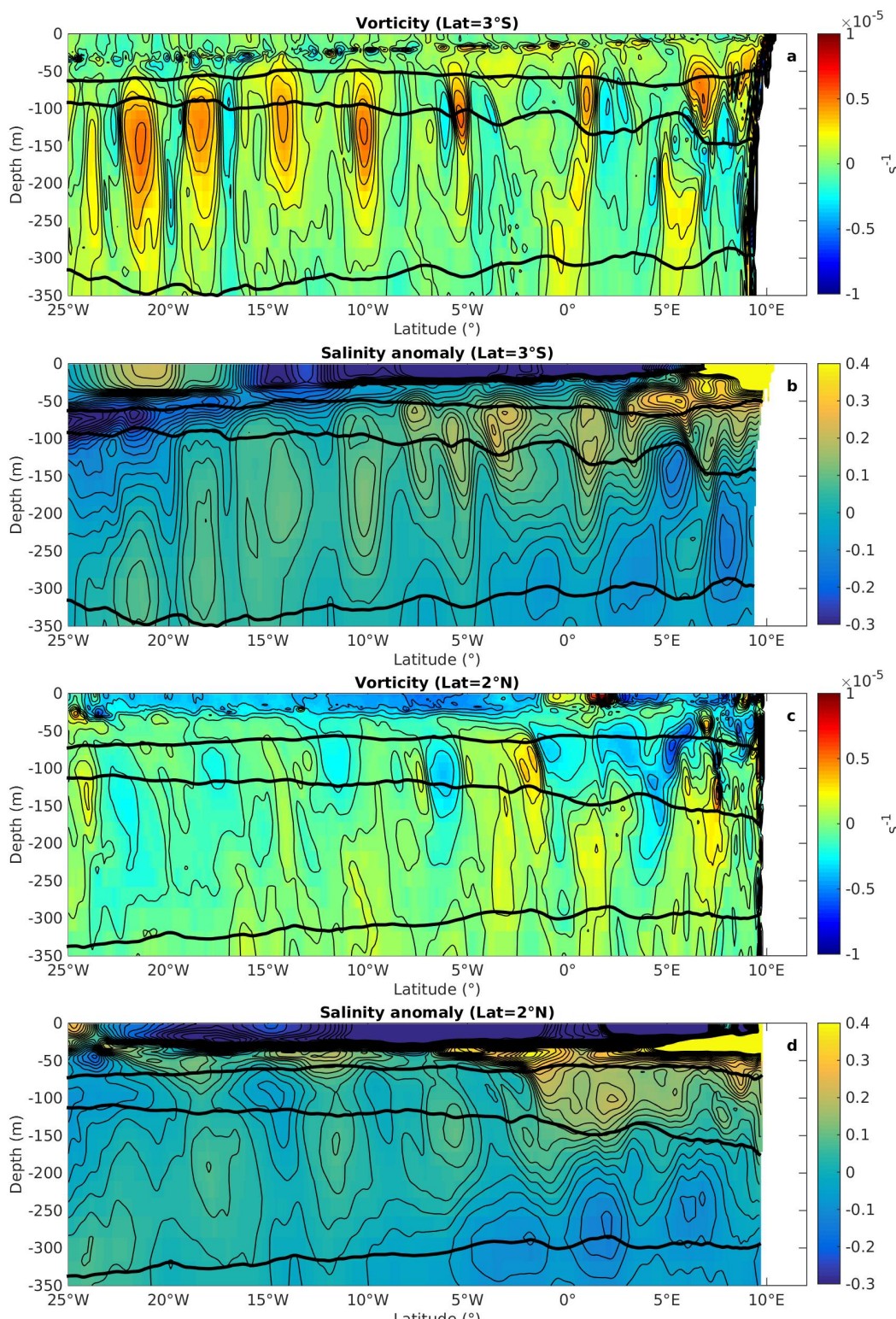

**Figure 11.** Zonal sections of vorticity and salinity along 3° S (**a**,**b**) and 2° N (**c**,**d**) on 10 May. Thick black lines represents the isopycnic levels defining layers 1 and 2.

### 3.3.2. Evolution of the Mesoscale Field

Figure 8 gives an idea of the time evolution of the isopycnic PV in each layer. Similar diagrams have been made in the northern hemisphere (2° N) and show that, in 2015, four to five vortices are formed north and south of the Equator. Their lifetimes can reach six months or more. For instance,

the anticyclonic vortex located at [0°, 3° S] in January (*Time* = 0 *day* and 0° in Figure 8) can be followed up to July (*Time* = 200 *days*) when it is finally stirred and dissipated near 20° W. Movie S1 (Supplementary Material) represents the daily evolution of PV and salinity in both layers and provides a general view over the whole equatorial Atlantic. Contours (in black) underlining the stronger PV cores have been added for a better readability.

The displacement of the vortices corresponds to the observed propagation speed of surface vortices at this latitude associated with the planetary beta effect [69]. Nonlinear vortices are indeed self propagating westward on the beta plane [70–73]. Other processes can also influence their dynamics and displacement speed [52,74–76], but westward self propagation by the planetary beta effect is often one of the major components [77], and we believe that it explains the trajectories of the anticyclonic eddies in our simulation.

While moving westward, vortices spread their PV and salinity content along their path, as can be seen from the filamentary structures ripped from their main core (see Movie S1, Supplementary Material, and Figure 7). These filaments horizontally mix with the surrounding waters, forming the large scale homogeneous PV and salinity tongues described above, associated with the westward and eastward (SEUC and NEUC/GUC) currents, at least up to 20° W. Further west, eddies interact with the turbulence and currents from the western region and are dissipated, especially in layer 1, whose PV tongue extends less westward (see also Figure 5). Their dissipation is also seasonal. Figure 8 shows that strong vortical signatures have disappeared after day 225, corresponding to the month of August. In fact, vortices are dissipated and their formation stops from late boreal spring to the end of summer. This can be seen more clearly in Movie S1 (supplementary material). The dissipation mechanisms will not be studied in detail here, but eddies undergo strong horizontal stirring and mixing, diffusing their salinity and PV contents within the large scale PV tongue. Their formation and westward propagation resume in September/October (day 300 or so) when a new cycle begins. Thus, the vortex lifetimes depend on their date of formation. If it can reach six months for vortices generated in autumn or winter, it is drastically reduced for vortices generated in spring. The interruption of the generation of vortices corresponds to the equatorial upwelling season. It is consistent with the observed vanishing of the EUC and the erosion of the high salinities in the Gulf of Guinea [14,24].

From their formation to their dissipation by isopycnic stirring and dissipation, the PV content of particles forming the core of vortices seems unaltered and drives the dynamics of the mesoscale vortices. They maintain the large scale PV tongues and associated mean circulation in the subsurface layers of the Gulf of Guinea. This part of the dynamics, from mesoscale to large scale, is consistent with the adiabatic turbulent cascade ruled by the conservation of both energy and PV. On a beta-plane, the inverse energy cascade (towards larger scales) is limited by the Rossby waves regime [78–81]. The anisotropy of the Rossby waves dispersion relation limits the meridional scale to the so-called Rhines scale, resulting in the formation of zonal jets [82]. Later studies have shown that dissipation also plays an essential role in the limitation of the inverse cascade and numerous idealized forced-dissipated simulations on a beta plane have shown the spontaneous emergence of zonal flows [82–85]. Once created, these zonal jets are very stable. One key process for their maintenance against dissipation is a positive feedback due to the radiation of Reynolds stress through a subtle interaction between waves, jets and turbulence, also called 'jet sharpening mechanism' [86,87]. The underlying physical process is the reorganization of the PV by the eddies which homogenize the background PV in their vicinity [79,80]. This creates a staircase profile where eastward jets are associated with strong PV gradients and westward jets with low PV gradients. The staircase profile and the high PV gradients also act as a waveguide for the waves and the eddies, maintaining this regime. This mechanism is consistent with the present simulation. The main difference is that, in the traditional Rhines cascade, PV is locally homogenized by eddies that are generated everywhere, and PV takes the planetary vorticity value, averaged over the Rhines scale, generating the staircase profile. In the present simulation, vortices are formed in a specific area with a specific PV content that is diluted along their path to form the PV tongue. Inside the PV tongue, PV is determined by the PV content of the vortices.

In our simulation, the generation of PV extrema forming the core of vortices is localized in the eastern part of the Gulf of Guinea, close to the African coast, by mechanisms we are now going to qualitatively describe.

### 3.4. Mechanisms for the Generation of PV Anomalies and Vortices

The strength of the PV anomaly associated with the core of anticyclonic vortices involved in the formation of the westward and eastward currents is $\delta PV \simeq 0.5\text{--}1 \times 10^{-5}\text{s}^{-1}$. We can distinguish different mechanisms able to generate potential vorticity anomalies (see [48] and Figure 12):

- adiabatic effects associated with meridional advection of particles within a varying background potential vorticity gradient (planetary $\beta$ effect and mean currents PV structure);
- mixing in boundary layers or within the water-column, mostly associated with strong vertical shear;
- frictional effects associated with surface wind or bottom drag, or lateral boundaries, in the case of vertical walls (which is the case when geopotential coordinates are used, such as in the NEMO model).

### Adiabatic process

*a/ Transport of PV (tracer) through background (meridional) gradient*

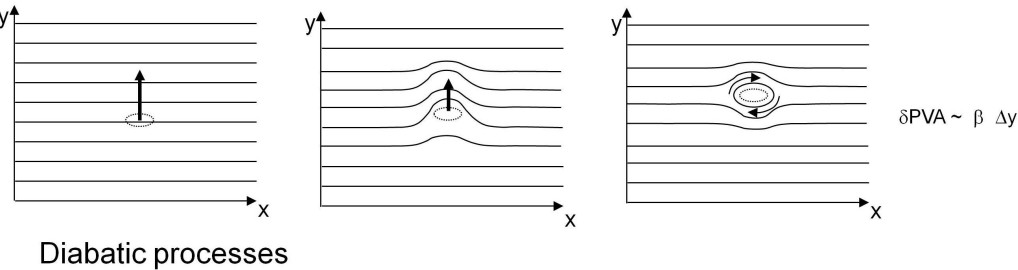

### Diabatic processes

*b/ Mixing of density field*                                      *c/ Friction*

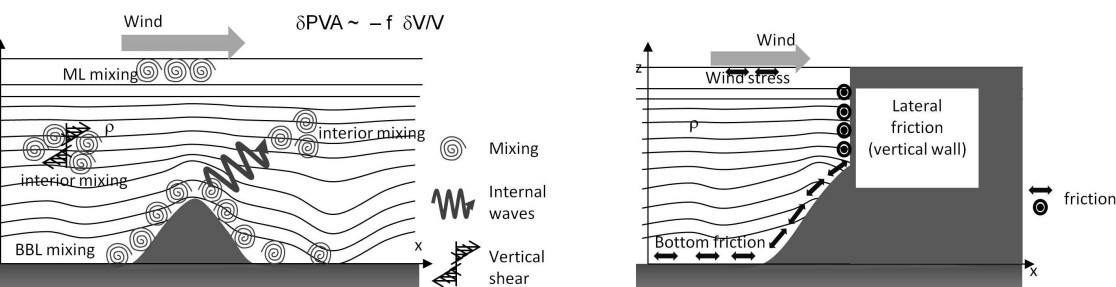

**Figure 12.** Illustration of the main mechanisms for the formation of PV anomalies and vortical structures. Adiabatic meridional advection generates PV anomalies when a particle is displaced within a background PV gradient (**a**). Diabatic mechanisms are associated with diapycnal mixing (**b**) or friction close to boundaries (**c**).

There can also exist diabatic and frictional effects associated with numerical approximations. They are thus sensitive to the spatial resolution used (in particular vertical resolution) and numerical choices (coordinate system and numerical schemes).

Finally, mixing or friction also modify the PV in layers situated above or below layers 1 and 2, which can lead to modification of the currents within layers 1 and 2. However, the current structure we examine exhibits maximums within layers 1 or 2, a signature that is necessarily associated with a PV anomaly located in these layers. In the following, we thus focus on local modification of PV within layers 1 and 2.

### 3.4.1. Advection

The generation of PV anomalies by adiabatic advection of a particle within a general PV gradient (Figure 12a) has been invoked to explain both the formation of the NEUC and SEUC [20]. It relies on the conservation of PV and creation of a PV anomaly by meridional displacement of particles, in turn generating a circulation. In a quiescent ocean where the PV gradient is associated with the planetary $\beta$ effect, a particle moving meridionally develops a PV anomaly given by

$$\delta PV \ = \ - \beta \, \delta Y \tag{2}$$

where $\delta Y = Y_{final} - Y_{initial}$ is the meridional displacement of the fluid parcels.

Taking $\beta \simeq 2 \times 10^{-11}$ m$^{-1} \cdot$ s$^{-1}$ and $\delta Y \simeq \pm 200$–300 km (assuming the particle is advected from the equator to 2–3° N or S) yields $\delta PV \simeq \mp 4$–$6 \times 10^{-6}$ s$^{-1}$, corresponding to the 0.5–$1 \times 10^{-5}$ s$^{-1}$ PV anomalies observed in the subsurface eddies. The anomaly would be negative in the Northern Hemisphere (particles with a low PV value moving northward to a higher background PV) and positive in the Southern one, yielding anticyclonic PV anomalies in both cases.

### 3.4.2. Mixing

Based on observations and salt budgets, Gouriou and Reverdin [10] showed that there exists strong mixing within the EUC and that mixing is a major component of its dynamics and variability in the eastern equatorial Atlantic.

Mixing is ubiquitous at the surface or bottom of the ocean where it is associated with convection or boundary layer turbulence. Below the pycnocline, layers have little or no contact with the surface or the bottom, but mixing can be generated by shear instabilities (Kelvin Helmholtz, symmetric, ...) or other processes (Figure 12b). Internal gravity wave breaking, double diffusion or other layering processes can play a role in the small scale structuring of tracers [88] but are not taken into account here. In particular, tides and internal tides are not represented in the simulation we analyse, but we believe they can induce significant mixing close to the shelf, which can in turn modify the PV evolution. This is further discussed in the concluding section.

The effect of diapycnal mixing on PV in general is controlled by the impermeability theorem [89,90]. This powerful principle shows that PV increases (a positive anomaly develops with respect to their initial value) in layers losing volume by mixing and PV decreases (a negative anomaly develops) in layers gaining volume. This principle has been shown to generate strong modification of the PV field, with both positive and negative PV anomalies, sometimes leading to the baroclinic destabilization of a current [49].

From [89,90] (see also [49]), the bulk modification of PV within a layer is given by

$$\delta PV \ = \ - f \frac{\delta V}{V} \tag{3}$$

where $\delta PV$ is the PV change in the layer where mixing occurs, $V$ is the initial volume of the region undergoing mixing and $\delta V$ is its modification associated with mixing.

PV is thus not a tracer when mixing is present: for regular tracers, such as salinity, the final value after mixing depends on values of tracers in adjacent layers. For instance, considering a volume $V$ of water in layer 2, having an initial salinity $S_i$ that mixes with a given volume $\delta_V$ of the shallower EUC water having a salinity $S_{EUC}$ in layer 1, the final salinity in layer 2 is:

$$S_f \ \simeq \ \frac{\delta V \, S_{EUC} + V \, S_i}{V + \delta V} \tag{4}$$

This yields an estimate of the volume variations, based on the modification of salinity

$$\frac{\delta V}{V} \simeq \frac{S_f - S_i}{S_{EUC} - S_f} \tag{5}$$

Considering $S_i = 35$, $S_f = 35.5$ and $S_{EUC} = 36.2$ (typical salinities in layers 1 and 2 in the eastern area of the Gulf of Guinea on Figure 7), we obtain $\delta V / V \simeq 0.7$. This volume change can then be used with Equation (3) to estimate that the modification of the bulk PV value of the volume of fluid by mixing is $\delta PV \simeq \pm 5 \times 10^{-6}$ s$^{-1}$. Note that, because the created PV anomaly depends on the local Coriolis parameter, the sign of the anomaly is negative in the Northern Hemisphere and positive in the Southern one, yielding anticyclonic PV anomalies in both cases. Mixing of EUC water with underlying waters can thus also yield significant PV modifications, able to explain the formation of anticyclonic vortices emerging in the simulation.

### 3.4.3. Friction

Diffusion of momentum within the water column does not modify the bulk integral of PV [89,90] and generally induces only isopycnic diffusion of PV (this can be seen in Movie S1). Viscosity could also generate compensating PV anomalies of opposite sign, in particular when isopycnic surfaces have steep slopes, but we have not observed such a phenomenon far from boundaries in the present simulation.

The strongest effects of friction are expected to be associated with boundaries where a net PV flux is possible. Surface wind stress or bottom drag (Figure 12c) can induce drastic modifications of PV [48,91–95] with important consequences on the dynamics. In particular, [96,97] have shown how friction associated with the wind stress are involved in the dynamics of TIW. The considered layers are located in the subsurface and wind-stress can not be involved in the modification of PV. However bottom friction can be involved in the transformation of PV in regions where the considered layers come into contact with the topography.

In the present configuration, the presence of vertical boundaries at all depths can induce lateral friction, associated with horizontal viscosity terms and lateral boundary conditions. Viscous boundary layers have been proposed as a mechanism for the generation of vorticity and eddies [49,98,99]. It is difficult to estimate the associated PV variation, as it depends on details of the boundary conditions (free slip, no slip, ...), viscosity operator and profile of the velocity near the coast, but we can expect that, in the present simulation, bottom and lateral friction can combine and generate significant PV anomalies.

To conclude, all mechanisms discussed in Section 3.4 are likely to be involved in the formation of PV anomalies and anticyclonic vortices.

### 3.5. Lagrangian Analysis

Several studies have analysed the PV evolution leading to the formation of vortices and identified which specific processes are involved in the formation of PV anomalies [94–96,100–103]. Such detailed analysis is based on the PV evolution equation, governed by adiabatic advection and diabatic terms (mixing and friction). In ocean circulation models, diabatic terms depend on diffusion coefficients given by specific parameterizations. Unfortunately, these non-standard diagnostics were not saved for our simulations meaning that our analysis remains qualitative.

However, a first order analysis can be done using Lagrangian particles. Indeed, for a particle that is followed along its trajectory, PV can only be modified by diabatic effects. We follow [96] and use a Lagrangian particle back-tracking method to identify adiabatic (i.e., PV conserving) and diabatic (i.e., PV altering) contributions to the PV anomalies of the vortices. Since previous studies have identified that mixing associated with shear instability is ubiquitous in the EUC [10,41,65–67], we also calculate the Richardson number

$$Ri = \frac{-g\partial_z \rho}{\rho_o \|\partial_z U\|^2}, \tag{6}$$

where $\rho$ is the potential density and $U$ the velocity. Given the position of velocity and density points of the C grid, the Richardson number is calculated with the same points and at the same location as the PV, when the divergence form is taken (see Figure 2 in [48]). Thus, PV changes occurring when the Richardson number is small suggest shear instability-driven mixing is involved in the PV evolution.

### 3.5.1. Tracking Algorithm

Different tracking algorithms have been developed (see [104] and references therein) with different properties. Tracking algorithms rely on spatial and temporal interpolations, sometimes offering the possibility of adding unresolved physics using stochastic variability. Small errors in particle trajectories, resulting from their determination from time-averaged and interpolated velocity fields, can result in changes in PV along particle trajectories (particularly where PV gradients are large) that are not neccessarily linked to non-conservative processes. Vertical interpolation of PV can also generate spurious signals, especially near the pycnocline. This is somewhat attenuated by the use of the rescaled PV, but not completely erased. We have thus chosen to develop a simple tracking algorithm, based on an explicit approach. The daily model outputs are interpolated hourly using a cubic spline. The spatial interpolation for all fields—except PV—is simply second order. To limit interpolation errors of the PV field, we follow the scheme proposed in [48] for the calculation of PV within a single "PV cell" (limited by tracer points) and consider PV to be spatially homogeneous within this cell. A rigorous validation of the Lagrangian PV evolution would be based on the comparison between PV tendency terms and PV evolution along trajectories [96,105], but this is not possible for the present simulation. Alternatively, [104] concluded that a good way to assess the accuracy of a diagnostic based on particle tracking is to analyse the convergence of that diagnostic when the number of particles increases. This is what we have done and we have verified that using 500 particles leads to stable statistics (two different sets of initial particle positions chosen to be randomly distributed within the vortex lead to the same diagnostics). The sampling frequency (1 day) of the velocity field used to compute the particle trajectories also seems sufficiently accurate for the present diagnostics since degrading it to two days did not significantly modify the statistics.

### 3.5.2. Lagrangian Diagnostics

Particles are seeded in several vortices and we computed their trajectories backwards from 10 May to 3 January. In the following, we present results for the vortex discussed above (see Figures 9 and 10). The results were roughly similar for other vortices and the observed differences are discussed in the final section.

We computed the backward particle 3D positions and, at each particle position, we estimate the salinity, temperature, potential density, potential vorticity, and local Richardson number.

The 500 particles are randomly distributed horizontally and vertically within the vortex core in layers 1 and 2. Because of the complex shape of the vortices, in particular when they have been recently formed, particles are seeded within 200 km from the eddy center and in vorticity regions higher than $1 \times 10^{-6}$ s$^{-1}$. Figure 13 represents the initial (3rd of January, in red) and final (10 May, in blue) values of salinity (a) and PV (b) as a function of density. The evolution of PV for all particles is also represented (c). The final scatter-plots are much tighter, in particular for PV. Final PV values are close to the mean value of $\overline{PV} \simeq -2.8 \times 10^{-6}$ s$^{-1}$ with a standard deviation of the PV within the vortex core is $\sigma_{PV} \simeq 10^{-6}$ s$^{-1}$. The local Coriolis parameter at the latitude of the vortex center is $f \simeq -7 \times 10^{-6}$ s$^{-1}$, so the vortex PV anomaly with respect to the background is $\delta PV \simeq +4 \times 10^{-6}$ s$^{-1}$ and associated with an anticyclonic circulation. The initial PV values cover a wide range. PV and salinity changes are strong in the [1025.7, 1026.4] kg $\cdot$ m$^{-3}$ density range, indicating the influence of diabatic effects, but they remain modest for higher densities. Figure 13c shows that the diabatic modification occurs during the first 80 days.

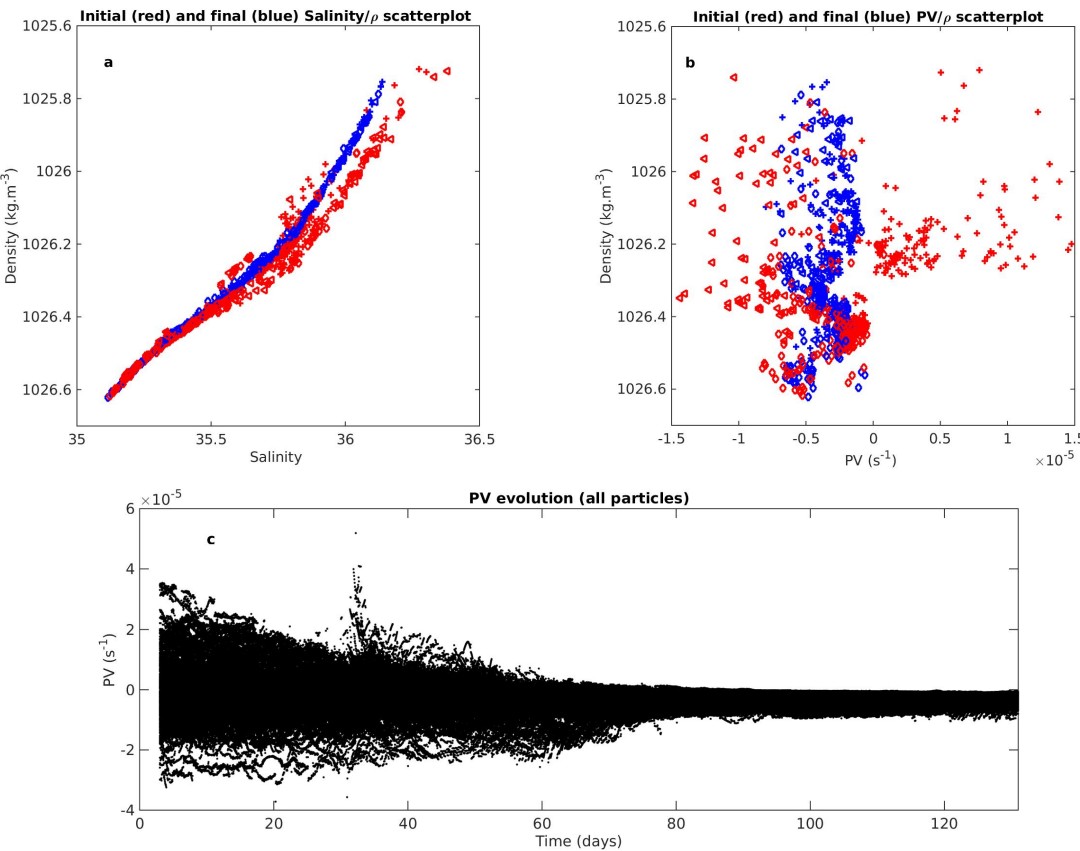

**Figure 13.** Initial (3rd of January, red color) and final (10 May, blue color) scatter-plot of salinity (**a**) and PV (**b**) against density. The ◇, +, ◁ symbols are associated with PV classes described in the text; (**c**): evolution of PV for all particles.

We then distinguish three classes of particles depending on the difference of the PV value between the final time (10 May, or day 131, when particles are in the vortex core) and at the beginning of the simulation (3 January, or day 3):

- Particles for which the initial and final PV value remain similar ($\|PV_{final} - PV_{init}\| < 2\,\sigma_{PV} = 2 \times 10^{-6}\ \mathrm{s}^{-1}$), identified with ◇ on the following plots.
- Particles characterized by relatively high initial PV values which undergo a strong decrease during their evolution ($PV_{init} > PV_{final} + 2\,\sigma_{PV}$), identified with +.
- Particles characterized by relatively weak initial PV values which undergo a sharp increase during their evolution ($PV_{init} < PV_{final} - 2\,\sigma_{PV}$), identified with ◁.

Figure 14 represents the initial and final position of all particles. The previous symbols, associated with colors, have been used to classify the particles, which come from many different areas and have different initial PV values. Note that most particles forming the vortex core are located north of the final position. Thus, southward displacement of particles plays a significant role in the formation of PV anomaly and of the anticyclonic vortex. However, some particles also undergo diabatic transformations and their PV is modified during the evolution. This determines the final PV of the particle and its PV anomaly when reaching the vortex core latitude. For instance, particles with low initial PV values moving to the latitude of the vortex would be associated with a negative PV anomaly and cyclonic circulation. Particles with initially high PV values would be associated with much stronger positive anomalies than observed, possibly generating much stronger anticyclonic vortices. For these particles, the transformation of their PV values by diabatic effects is crucial to explain the observed final PV content.

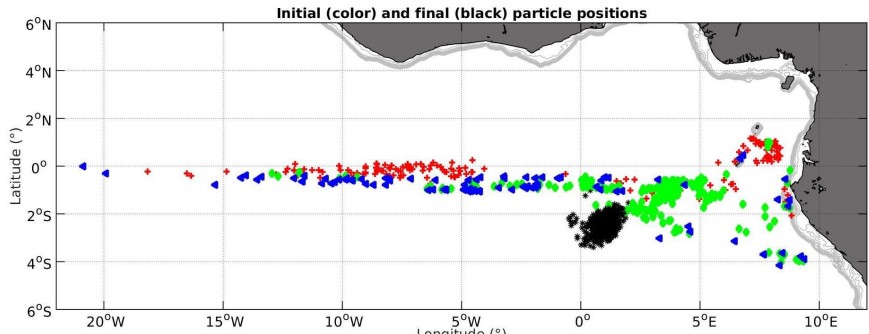

**Figure 14.** Initial (3 January = day 3) and final (10 May = day 131) positions of all particles constituting the vortex core. The initial position is identified by the colored symbols red + for particles with high initial PV values, blue ◁ for particles with low initial PV values and green ◇ for particles with weak variations between initial and final PV values. The final positions are identified by black ∗. The thick black line represents the coastline of the model and thin gray lines are isolines of the water depth, from 50 to 500 m.

Most particles undergoing moderate PV transformation during the 128 days evolution are located in the vicinity of the final vortex position, close to the Equator and on the eastern side of the Gulf of Guinea. Particles with low initial PV mainly come from a zonal band, south of the Equator and extending west (0–20° W). A few are located south-east of the final position. Particles with high initial PV come from two major areas, close to the Equator, but north of and less spread than the low PV particles (5–15° W), and north-east of the final position, between Sao Tome and Principe islands. A movie (Movie S2 in supplementary material) is proposed to give a more precise picture of the dynamics that is described below, in particular the grouping of particles from different regions and classes in the vortex core.

Figure 15 represents the trajectories of initially low PV particles, as well as the evolution of their density and PV values and Richardson number (we only represent values for $Ri < 1$ to identify the possibility of shear-instability-driven mixing). We distinguish two groups of particles depending on their initial position: along the Equator (panels a–c) or in the south-eastern region near the African coast (panels d–f). The particles with initially low PV located along the Equator represent about 16% of the particles eventually forming the vortex core. Their main density range is $\rho \in [1025.9, 1026.4]$ kg $\cdot$ m$^{-3}$. During the first 80 days, they are first rapidly advected eastward, along $\simeq 1°$ S, they reach the eastern Atlantic and move southward. During this time period, their PV values are drastically increased. Note this also corresponds to a period during which $Ri$ is low (below 1) along many of the particles trajectories (compare panels b and c), indicating possible diapycnal mixing. After 80 days, they start to rotate anticyclonically, joining the low PV particles initially located in south-eastern region but with higher densities ($\rho \in [1026.4, 1026.7]$ kg $\cdot$ m$^{-3}$) and entrain them into the vortex core. The latter particles (lower panels) represent about 2.5% of the vortex content and are negligible.

Figure 16 is similar to Figure 15 but for particles with initially high PV. The particles with initially high PV and located along the Equator (a–c) represent 19% of the vortex content. Thus, the equatorial region provides similar amounts of low and high PV particles to the vortex core. They are also advected eastward by the EUC. Their density range is mostly $\rho \in [1026, 1026.3]$ kg $\cdot$ m$^{-3}$. Their PV values are drastically reduced during the first 80 days of the evolution after which they move southward, starting to rotate anticyclonically. Again, low $Ri$ are reached during this initial phase, indicating possible diapycnal mixing. High PV particles initially located northeast of the final position (d–f) have a similar density range and fate. They represent about 18% of the vortex content. They move slowly southward, mix and change their PV, and finally merge with other high PV particles advected by the EUC. High PV particles are essential in the formation of the anticyclonic vortex. They are initially associated with the highest PV anomalies and launch the initial rotation of the high PV particles (see Movie S2, Supplementary Material).

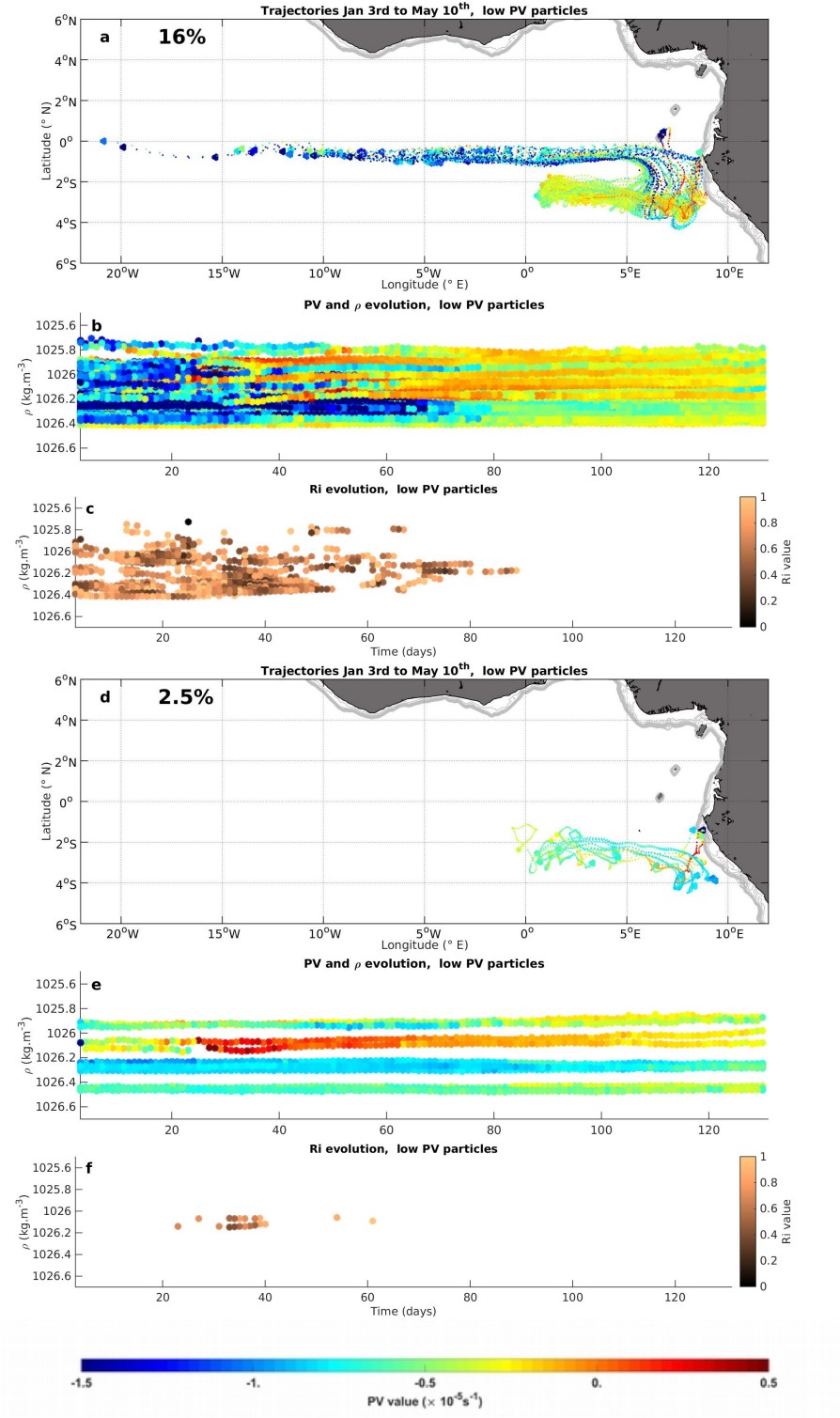

**Figure 15.** Trajectories (**a**,**d**), evolution of the density (**b**,**e**) and of the Richardson number (**c**,**f**) of low PV particles, eventually constituting part of the vortex core. The initial position is identified by the symbols ◁ and the final positions by ∗ on trajectories. Only values below 1 are represented for the Richardson number (indicating the possibility of shear-instability-driven mixing). (**a**–**c**): particles with low PV values initially located along the Equator. The thin gray lines are isolines of the water depth of the model, from 50 to 500 m; (**d**–**f**): particles with low PV values initially located south-east of the final position. For (**a**,**b**,**d**,**e**), the color corresponds to the PV value during the evolution (common colorscale given at the bottom).

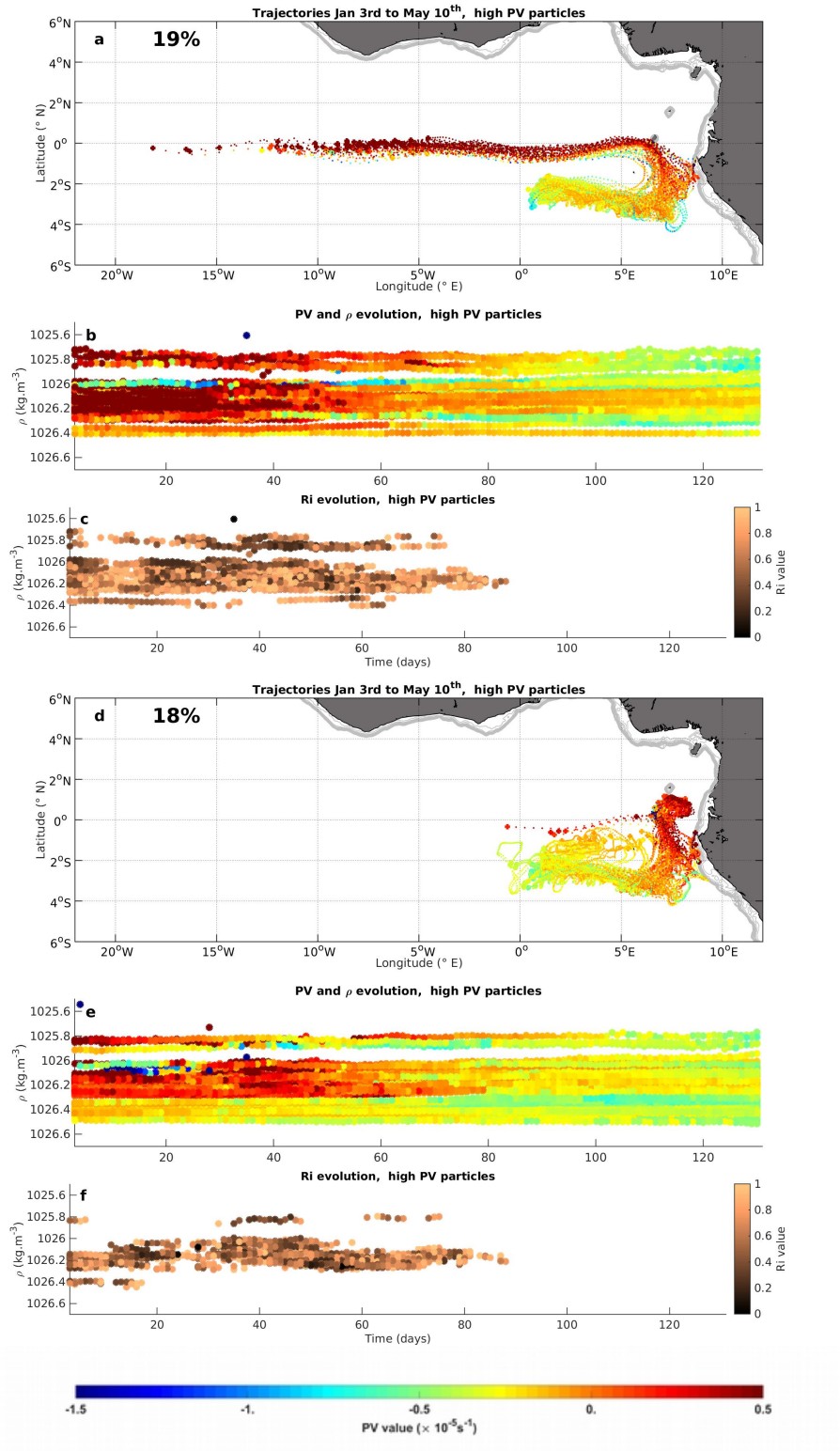

**Figure 16.** Same as Figure 15 but for high PV particles. The initial position is identified by the symbols
+ and the final positions by ∗ on trajectories; (**a**–**c**): particles with high PV values initially located
along the Equator; (**d**–**f**): particles with high PV values initially located north-east of the final position,
between Sao Tome and Principe islands. For (**a**,**b**,**d**,**e**), the color corresponds to the PV value during the
evolution and the colorscale is the same as in Figure 15.

Most of the particles with moderate PV change are initially located close to the final vortex position (see Figure 17a–c). They represent 25.5% of the particles eventually forming the vortex core. Other particles with moderate PV change come from the equatorial region or African coast (d–f). They represent 19% of the particles. They cover the highest density range forming the vortex core ([1026.3, 1026.5] kg · m$^{-3}$). They generally start their anticyclonic rotation later than the other particles discussed above, showing that they are mostly entrained into the vortex core by the eddy formed earlier by the particles from the high PV class (see Movie S2). For most of the particles with moderate PV changes, *Ri* remains above 1, suggesting that the impacts of diapycnal mixing are small. Some particles exhibit some strong PV transient change during their evolution (but eventually reach average PV values, which is why they have not been identified as high or low PV particles). This PV change is either associated with low *Ri* or is observed for particles passing close to the African coast or Sao Tome and Principe islands (Figure 17a,d). Modification of PV close to the topography is also visible for some high and low PV particles (see Figure 15a,d). This suggests that bottom or lateral friction play a role in the transformation of PV for some particles. We have however not been able to quantify this process.

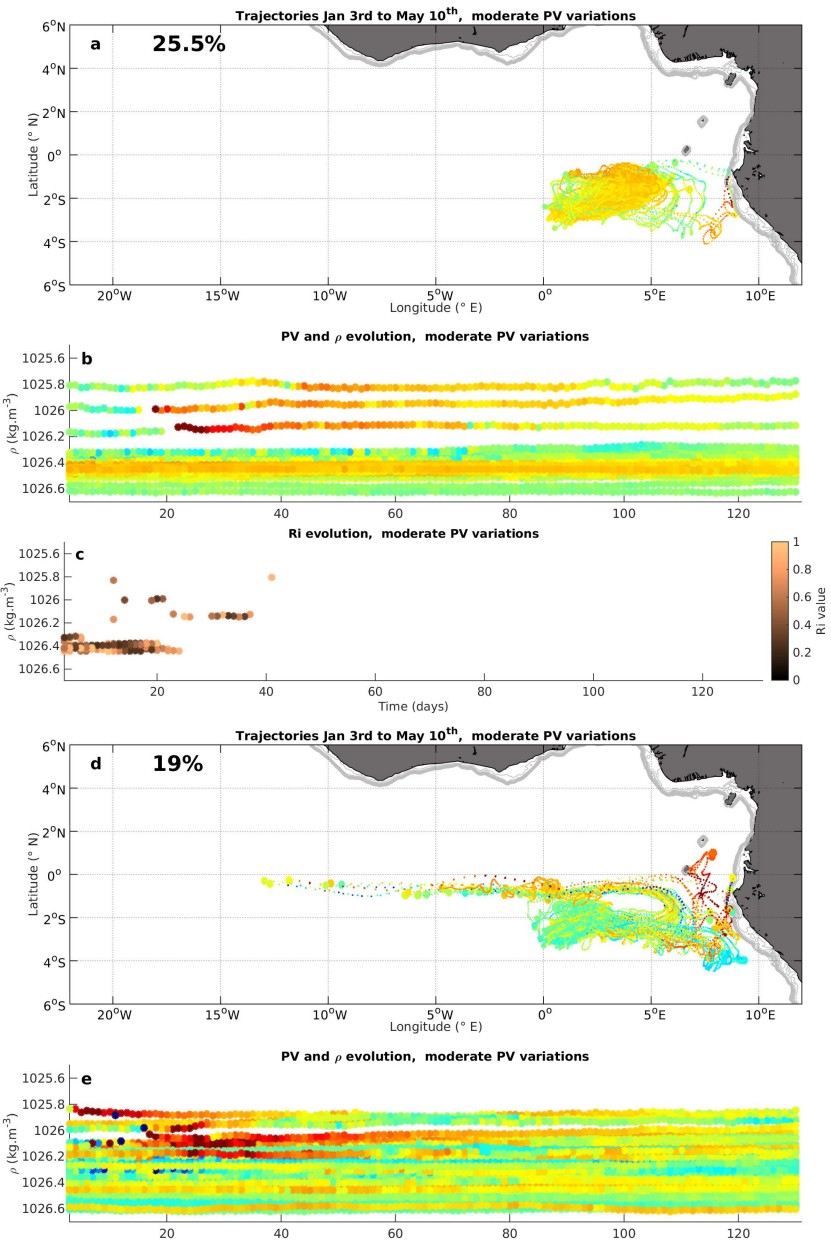

**Figure 17.** *Cont.*

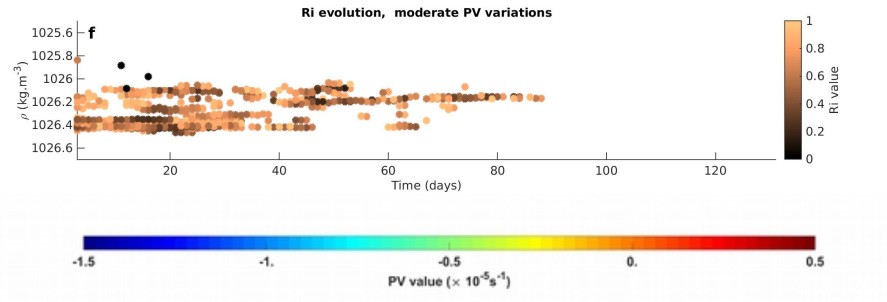

**Figure 17.** The same as Figure 15 but for particles undergoing a moderate PV change during the evolution; (**a**–**c**): particles initially located close to the final position; (**d**–**f**): other particles, coming from along the Equator or African coast.

To conclude, the qualitative analysis presented in this section shows that both adiabatic and diabatic processes contribute to the determination of the PV anomaly constituting the core of subsurface vortices propagating on both sides of the Equator. For the vortex we considered, diabatic effects transform PV during the early stages of the evolution by mixing or by friction when particles reach the continental slope of the African coast or of Sao Tome and Principe islands. For instance, zooming on these areas in Figure 7 (see also Movie S2) shows that PV filaments are formed in the vicinity of these regions. Unfortunately, the present diagnostics do not permit to precisely separate mixing and friction and evaluate their overall quantitative contribution.

Finally, as seen from Movie S2, particles forming the vortex core originate from different regions and are subject to different diabatic and adiabatic processes during their evolution, determining their PV values and anomalies. They then get in proximity and finally merge or align vertically forming a mesoscale structure. Merging is ubiquitous for surface vortices [106,107] and has been observed in situ for subsurface eddies too [108]. The present results show that, in our simulations, merging and alignment of eddies could be a major mechanism for the generation of large long-lived mesoscale vortices in the Gulf of Guinea.

## 4. Summary, Discussion and Perspectives

### 4.1. Summary

In this paper, we have analysed the ocean dynamics in the subsurface layer of the Gulf of Guinea, from large scale circulation to submesoscale. This yields a general story involving upscaling dynamics: first small scale diabatic (mixing/friction) and adiabatic processes lead to the generation of submesoscale structures; the latter merge and align vertically when they have similar anomalies forming the cores of subsurface mesoscale nonlinear vortices; nonlinear vortices propagate westward, spreading their PV and salinity anomalies with surrounding waters by stirring, filamentation and isopycnal diffusion. This leads to the formation of large scale homogeneous PV tongues and their associated circulation, forming the eastward and westward subsurface currents.

Downscaling processes are also important. They are involved in the formation of the large scale PV tongue (stirring and filamentation of mesoscale eddies) and mixing. It is also the structuration by the large scale circulation that leads to mixing in selected density ranges and areas, creating local PV anomalies. In both upscaling and downscaling phases, submesoscale processes are essential catalysts between large scale and mesoscale.

The observed spatial or temporal modulation of the zonal current system is associated with the seasonal formation cycle (December to June) of the eddies and vortices. The velocity signature of vortices superimposes on the larger scale circulation and induces strong spatial variations at mesoscale. Eddies are formed during the seasonal relaxation of the equatorial upwelling in the eastern equatorial Atlantic Ocean. In our simulation, anticyclonic vortices are generated on both sides of the Equator

and explain the formation and maintenance of the westward and eastward (SEUC and NEUC/GUC) currents east of 20° W.

The analysis proposed here relies on PV, which is the key quantity to analyse geostrophic circulation, vortices and up/downscaling physics. It also allows for identifying diabatic events so that analysis of the evolution of PV, following particles or within layers determined by isopycnic surface, allows for inferring the respective influence of adiabatic and diabatic processes on the circulation. The PV diagnostics used in this study have been developed for the NEMO code and are available upon request.

### 4.2. Discussion

Different mechanisms have previously been proposed to explain the formation of zonal subsurface equatorial currents and in particular the eastward Tsuchiya currents [109]. The mechanism discussed here is related to the beta-plume dynamics [110–112] for which local diabatic forcings modify the PV structure and establish a gyre westward of the forcing region. This mechanism has been invoked in previous studies to explain the formation of the Tsuchiya jets [113–115] (or other zonal current systems, [116]). In these studies, mixing, associated with equatorial upwelling, has been recognized as a driver for the formation and maintenance of the jets.

However, the present numerical results bring new insights and interpretations. The PV tongue forming the jet system is established by anticyclonic vortices transporting and diluting their PV content along their path. The eastward Tsuchiya jets and the westward currents, observed on their equatorial flanks, are part of the same system. The anticyclonic vortices undergo strong meridional displacements in the early stage of their life, so the PV tongues and associated basin scale gyres are not located westward of the forcing regions. The presence of vortices, and the strong seasonal cycle of their life cycle, induce strong temporal and spatial modulations of the zonal currents. Such a nonlinear beta-plume regime has been originally discussed in [117] for simplified configurations.

Even though we could not identify their respective quantitative contribution, different adiabatic or diabatic forcings are likely to play a role in the generation of PV anomalies: meridional advection, vertical mixing and friction. Interestingly, many previous studies have focused on one of them to study the formation of the Tsuchiya jets, for instance:

- meridional advection was mentioned in [20,118–121] and in particular in [122] who associated it with transport by nonlinear eddies. Lateral diffusion of vorticity from the EUC, generated by horizontal diffusion, was also the mechanism proposed in [123] and was interpreted as representing meridional advection of PV by mesoscale eddies;
- mixing and/or a source of anomalous water masses (sometimes parameterized by a relaxation term) was the driver in many studies [113–115,119–121].

In our simulation, all forcing mechanisms could equally contribute to the formation of small scale PV anomalies in the eastern equatorial region. The PV perspective shows that, whatever the mechanism for the generation of PV anomalies, the small scale PV cores eventually merge and align to form mesoscale anticyclonic vortices transporting their PV content to form the large scale PV tongue. Thus, the approach we proposed here accredits and integrates all studies mentioned above.

Some of the mechanisms we discussed are also typical features of the inverse cascade of geostrophic turbulence on the beta-plane, with some differences with the traditional Rhines cascade [78,81,82,86,87]: here, the regions of homogenized PV are formed by vortices emerging from a small area; inside the PV tongue, PV is determined by the initial vortices and does not correspond to an average of the planetary vorticity over the Rhines scale.

Rossby waves, instead of nonlinear eddies, have been shown to play a significant role in the formation of mean jet-like currents in some studies (see [109] and references therein). Rossby waves can be important to explain other jet-like structures observed in the present simulation, in particular, the western region or deeper low latitude intermediate currents [23,44,45,124].

Another striking feature that we have not discussed is that the mean PV profiles exhibit a change in sign of their gradients (see Figure 6). The Charney–Stern [125] condition for instability is met, but no eddies seem to emerge from the mean zonal current. In fact, the mean currents and PV fields being temporal average, their structure is a mixed combinations of the time average signature of the train of vortices generated in the eastern part of the basin and propagating westward and of a permanent large scale PV structure (which is itself influence by the vortices). Presumably, the latter permanent PV signal has a weak anomaly, without change of PV gradient, while vortices have a strong anomaly which leads to changes in PV gradients in instantaneous fields (and of the total mean PV structure if the signature of vortices dominates), without generating barotropic instability.

Thus, it is possible to have mean (time average) PV fields like observed, and no instability, if the vortex signature dominates the mean signal.

Concerning the formation of anticyclonic vortices, we have focused on one vortex in the Southern Hemisphere. Other vortices, in particular from the Northern Hemisphere, have been analysed. Meridional displacement mixing (associated with low Richardson numbers) and friction play a role on the formation of southern and northern anticyclonic vortices. The contribution of each mechanism can of course vary and the fact that the vertical structure of vortices, and in particular the position of their cores, varies significantly (see Figure 11) could be attributed to their generation mechanism. Indeed, merging and alignment of smaller, submesoscale, structures generated in different areas and density ranges by diabatic and adiabatic processes is a random process that can explain the variability of the vortices that are generated.

The diagnostics derived from the Lagrangian analysis remain qualitative and are validated by their stability when a large enough number of particles is used (here 500). Another validation is also the general coherence between different physical diagnostics: particles that undergo PV changes due to mixing mostly come from the EUC where mixing is prominent and where salinity is also modified; PV modification mostly occurs during a time period when Ri is small or when particles reach the continental slope. They were used to present evidence that both adiabatic and diabatic processes are involved in the formation of the vortex. However, we have not been able to quantify their relative contributions in more detail. A more rigorous validation or the use of a more sophisticated Lagrangian tracking method [104] can be necessary for quantitative diagnostics. For instance, near the coast the flow is likely characterized by strong velocity and tracer gradients, and we may expect large potential errors in the Lagrangian diagnostics. Thus, it remains unclear whether the PV changes we observe for some particles are significant or not. The daily model outputs was sufficient here, but for configurations with higher resolution, higher output frequencies could be necessary, in particular if tides and internal tides are represented.

This study underlines that submesoscale processes are major ingredients of the mesoscale to large scale circulation: they define the PV value of particles which in turn determines the dynamics at geostrophic scales. In ocean circulation models, part of the submesoscale dynamics (mixing, friction and isopycnal diffusion) relies on parameterizations and numerical choices that are questionable. In our simulation, this is the case of lateral friction for instance, which is an inaccurate representation of bottom friction. Distinguishing the contribution of a specific mechanism requires sophisticated online diagnostics [94–96] and was not possible here, so we can not evaluate the importance of lateral friction in our results. However, this study confirms that, in circulation models, parameterizations modify the PV field and shape the dynamical structure of the mesoscale to large scale circulation. Thus, the processes identified here remain associated with a numerical simulation and are not necessarily realistic.

### 4.3. Perspectives

In this study, we have focused on the eastern area and analysed a simulation over one year. Given the intensification of some currents in the western area, other processes are undoubtedly involved in the strengthening, and possibly the formation, of subsurface equatorial currents in this

region, as proposed by several studies [3,4,20,22]. The interaction and continuity/discontinuity of the eastern and western subsurface jet systems is of major interest. Some studies have in particular noticed some changes in the isopycnic and meridional positions of the NEUC and SEUC currents from west to east, the main cores of the currents intensifying in lighter density range and moving slightly poleward in the eastern basin [43,118]. Here, this applies the Atlantic SEUC and could be associated with the existence of eastern and western systems, which can merge in the middle of the basin. Alternatively, in the eastern basin, interaction with the intense circulation of upper layers could generate a slow erosion of the upper part of the mesoscale vortices as they propagate westward, a mechanism that could explain more active cores in lower layer, as they move westward. In addition, anticyclonic vortices also propagate Equatorward [70–73], which could explain the Equatorward displacement of the jets from East to West.

In the Atlantic, the African continent introduces an interesting equatorial asymmetry. Contrary to the SEUC, the NEUC is only found in the western part of the basin, as, given its poleward shift, the African coast prevents it from penetrating further into the Gulf of Guinea. In the eastern basin, it is replaced by the GUC. Both currents are usually seen as disconnected [1,11,24], but our simulations clearly show that the mesoscale vortices involved in the formation of the GUC, and the PV signature of the GUC extends far west in layer 1, at least up to 20° or even 25° W. This leads to a possible interaction with the NEUC which is worth studying.

A serious limit of the present result is that it is difficult to distinguish if there is a dominant process from the qualitative estimates presented above. Quantitative analysis, using PV tendency terms, are necessary to refine the results presented here and identify the major mechanisms determining the PV structure of the core of vortices generated in the Gulf of Guinea. This requires saving PV tendency terms, which we think would be a valuable effort for realistic models in general. Moreover, since diabatic submesoscale processes are so important for the cascade presented here, it is certainly worth investigating the influence of tides. Strong internal tides generated at the shelf break may increase the impact of diapycnal mixing [126]. Interestingly, recent work ([127] see their Figure 4) have analysed the specific influence of different mixing processes on the downward flux of heat within the pycnocline, in a global 1/4° model. In the Gulf of Guinea, mixing due to shear instability is the most important mechanisms for heat transfer along the Equator and mixing due to parameterized internal tides dominates along the continental shelf break. Even though their local impact is moderate, boundary layer and other (background) effect also contribute to mixing. The analysis for PV, including frictional processes, remains to be done.

The seasonal variability of the EUC dynamics is associated with the equatorial upwelling season and is well known [11,13,14,24,41]. This acts on the diapycnal mixing within the EUC but more generally in the vicinity of the Equator. During this season, the mesoscale vortices are rapidly eroded, diluting their PV content and disappearing. The details of the erosion mechanism, and its impact on the westward extent of the PV tongue, are worth investigating too.

Here, we focused on year 2015. Mixing within the pycnocline being involved in the processes we studied, interannual variability of the surface layer and of the atmospheric forcings can lead to modifications of the present results. In the Gulf of Guinea, the seasonal variability generally dominates [128], so we do not expect strong modification of our results for usual years. However, stronger modifications are observed after a Pacific El Niño. It would thus be interesting to redo the present study for 2016. In fact, it would be interesting to combine such an interannual variability study with the more quantitative analysis of PV dynamics discussed above, so as to identify which processes are responsible for the modification of the subsurface turbulent cascade we exposed here.

The processes studied here are associated with numerical models. Even though they offer an interesting and possible mechanism for the generation of subsurface zonal jets, observations are needed to confirm their realism. In particular, equatorial bands and the Gulf of Guinea are known to have few surface vortices [68,69]. The present simulation reveals a very rich and intense mesoscale activity below the pycnocline, with large anticyclonic vortices propagating westward for more than 100 days.

This has to be confirmed by high resolution observations. For instance, longitudinal velocity and salinity sections along 3° S or 2° N could help identify local salinity maximum, and latitudinal sections could confirm it is indeed associated with a vortical structure.

Finally, the mechanisms presented here depend on the processes involved in mixing (Kelvin–Helmholtz, symmetric or inertial instabilities for instance) and on their parameterizations in circulation models. Comparison with other models is thus also a valuable perspective.

**Supplementary Materials:** The following are available online at Available online: https://www.mdpi.com/2311-5521/5/3/147/s1, Movie S1: PV SAL 2layers new; Movie S2: Trajectories new.

**Author Contributions:** Conceptualization, Y.M.; methodology, Y.M., J.J., R.H, C.M. and A.K.-L.; data curation, J.J.; software, Y.M. and F.A.; validation, F.A., S.C., F.M., I.D., A.C., G.A., B.B. and Y.M.; formal analysis, Y.M., F.A., A.D., M.A., R.H. and A.C.; investigation, F.A. and Y.M.; resources, Y.M.; writing–original draft preparation, Y.M.; writing–review and editing, all co-authors; visualization, F.A., Y.M., A.C., A.D. and R.H.; supervision, Y.M.; project administration, Y.M.; funding acquisition, Y.M. and A.C. All authors have read and agreed to the published version of the manuscript.

**Funding:** This research was funded by CNES (project Alti-ETAO), IRD (program JEAI-SAFUME) and is a contribution to the TriAtlas project funded by European Union's Horizon 2020 research and innovation program (Grant No. 817578). Supercomputing facilities were provided by CINES and GENCI (project GEN7298) and by CALMIP (project P18033).

**Acknowledgments:** Leif Thomas and two anonymous reviewers provided useful comments that helped improve the present manuscript.

**Conflicts of Interest:** The authors declare no conflict of interest.

## Abbreviations

The following abbreviations are used in this manuscript:

EUC     Equatorial Undercurrent
NEUC    North Equatorial Undercurrent
SEUC    South Equatorial Undercurrent
GUC     Guinea Undercurrent
SEC     South Equatorial Current
NEC     North Equatorial Current

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
