# Peer review of "From Mixing to the Large Scale Circulation: How the Inverse Cascade Is Involved in the Formation of the Subsurface Currents in the Gulf of Guinea"

_fluids, doi:10.3390/fluids5030147_

Round 1

Reviewer 1 Report

Review of
From mixing to the basin scale circulation: how the inverse cascade is involved in the formation of the subsurface currents in the equatorial Atlantic

by
Fernand Assene, Yves Morel, Audrey Delpech, Micael Aguedjou, Julien Jouanno,
Sophie Cravatte, Frédéric Marin, Claire Menesguen, Alexis Chaigneau, Isabelle Dadou, Gael Alory, Ryan Holmes, Bernard Bourlès, and Ariane Koch-Larrouy

Summary: 
Assene et al. perform high resolution realistic numerical simulations of the
equatorial Atlantic ocean to study the circulation in that region. The authors
describe the resulting currents with good attention to detail, and analyze the
currents and vortices using potential vorticity.

Their work is interesting and well-presented.  I have only a few suggestions that could potentially strengthen the results and significance of the research.  I am recommending that the journal Accept after minor revision.  

Detailed comments and questions directed to the Authors:  

The simulation of the equatorial Atlantic circulation that you present is very nice, and the detail with which you describe the currents is also good.  My only significant criticism has to do with how PV is used in the analysis. 

Main Suggestion

1. Although you compute PV, you don't compute any estimates of the various terms that affect the changes in PV, like advection, diffusion, and mixing.  As a consequence, you aren't able to assess which of the PV forcing mechanisms are the most important ones.  If you were able to come up with reasonable estimates of PV forcing terms, and demonstrate that the various terms account for most of the PV changes you observe, you could then address the question of what is causing the change in PV. That would be a very significant improvement to the paper.  

Smaller Suggestions and Questions

2. What was the wind field during the year analyzed?  Are the surface currents directly driven by the surface wind stress?  I wonder if the momentum flux at the surface could be a source or sink of PV?   

3. You performed a simulation over the years 1993 - 2015. How much of that was necessary for spin-up? Do you have a metric (region-integrated KE, for example) that you used to identify when the sims reached steady-state? With such a long run, this is a great opportunity to do a longer time-average and perhaps study deviation-from-average, or even define what a typical year's cycle of currents looks like.  Said another way, is there a way to convey quantitatively how typical the 2015 circulation is?  

4. You refer to Figure 4 as evidence of PV homogenization, but it is difficult to see in a color plot.  It would be easier to see the PV homogenization in a line plot of PV vs latitude.  The PV would have to be averaged over some longitude range, or some typical longitude would have to be selected.  

5. Section 3.4 discusses the mechanisms of PV anomaly generation, but the discussion is hypothetical at times in the sense that it does not take advantage of the simulations themselves. I think Fig. 10 is great, but some quantitative use of the numerical simulations to support the ideas in this section would be a significant improvement.  (This is a reiteration of the Main Suggestion above.)

6. In Section 3.5 (Lagrangian analysis) I like your idea of tracking the particles backward from a specific eddy. It is interesting to see that the fluid that eventually makes its way into the eddy comes from a variety of different places. However, I don't find the PV analysis that helpful. The PV of the various parcels seem to be pretty much correlated with latitude of wherever they are at the moment, and it is not clear that PV conservation plays a significant constraint or effect on the flow.  This is again related to the Main Suggestion.  

7. In Figs. 13-15, I really like the identifying the days that certain particles have low Richardson number, but it is not clear to me that those days are correlated with PV changes.  Is there a way to quantify whether low Richardson number parcels typically undergo a change in PV?  Also, it would help the reader if you could detail how you computed Richardson number in this numerical setting.  

Typos

8. You did a very nice job with the polished detail of the manuscript.  I could only find a few typos.
Line 87, -1 superscript in the units, 2 locations
Line 130, 8.3 E - 1 N should the dash be a comma?
Line 271 component -> components
Line 348 loosing -> losing
Line 365 surface -> surfaces
Line 394 signal -> signals
Line 399 consider PV is -> consider PV to be
Line 450 anoamlies -> anomalies
Line 578 regions of homogenized PV is formed -> regions of homogenized PV are formed
Line 638 formatting on degree symbol 25 W looks like capital O instead of number 0.
Caption of Figure 1 (3rd line from bottom):  The bottom left panel represent -> The bottom right panel represents

Reviewer 2 Report

Review Manuscript 879489-FLUID

In the manuscript 879489 entitled “From mixing to the basin scale circulation : how the inverse cascade is involved in the formation of the subsurface currents in the equatorial Atlantic”, Assane and co-authors analyze how the subscale processes (aka mixing, friction and filamentation), connect with mesoscale vortices and influence the circulation of the equatorial Atlantic Ocean, based on the potential vorticity (PV). The work uses a high resolution numerical model as an experimental basis, focusing on a specific period of real time. This is a dense and well-produced job. However, the authors based their study on a specific period of time as being representative, without considering climatic events exogenous to the processes studied, and which may influence the results (see comments). Even if it does not change the result, some consideration should be raised in the manuscript. The manuscript also changes its central line which begins by speaking the Tropical Atlantic and changing dramatically to the Gulf of Guinea. The suggestion is to make the central line of the article more defined. Another central comment is that interpretation based on snapshots or individual events over time (although it is a valid methodology) is questionable for the impact of the conclusions. Perhaps a small rearrangement of the text and inclusion of some observational framework will contribute to the “take home message” of the article (see comments). I suggest some adjustments to the text and rework of the figures, believing the article will be much more attractive to the public.

In general, some figures use the same color map for two different fields, which makes interpretation difficult with the text. See specific comments on the figures and suggested changes.

In this sense, I recommend the manuscript for publication with major reviews, mainly in the figures and discussion. 

--- specific comments ---

Line 6 “Guinea UnderCurrent (GUC)”... insert space

Line 23 … “the subsurface oxygen minimum zone” … (only in the eastern Atlantic margin)

Line 40 Please explain better this sentence “Particles flow along the Brazilian coast before entering the EUC”.

Lines 63-64 “We first show that the characteristics of the subsurface currents are coherent with observations (section 3.1).” The authors argue a lot that the results are close to the observations, however, I did not see the result of any observation except other articles in reference form.

Line 77 “Tides are not taken into account.” Can the authors rule out the influence of tides in diapycnal mixing ? (any ref or work previously done ? And what about the model's bathymetry?)

Lines 101-103. neutral between 1993-2015? Check the variability of the 2015 Tropical Atlantic. 2015 was a relatively strong year for El Nino, with impact on the Atlantic Dipole and mainly on ITCZ. Authors should reinforce why they chose 2015 as the representative year. What really concerned me is the question of the particular year they chose - 2015 (Why did not to use a climatology, excluding the years with positive and negative indices, could you exclude the time series index? Was this done?). 2015 was a year of a strong El Nino (the second strongest in 70 years), but we know both TNAI and TSAI were neutral in 2015, which is a surprise. TSAI was "on edge" to consider itself positive. You need to see if it has any influence, even though it is a study about subsurface currents. Is there any lack of ocean-atmosphere correlation and interaction in this case ? This is a very peculiar and active area of ​​interaction, the variability of the Tropical Atlantic is basically modulating the evolution of the African Monsoon -> like the pressure seesaw system on the continent of the ocean, due to the geometry of the Tropical Atlantic basin. Another very strong influence is the ITCZ's southern variation, as well as more external characteristics such as NAO, SAM and ENSO itself. Unfortunately, these aspects were not even mentioned in the text, even in a supplementary analysis.

Lines 130-131 “In our case, the profile used for the rescaled PV is located at 8.3oE − 1oN on the 10th of May. It represents the typical stratification of the Gulf of Guinea where vortices are formed”. Any reference to prove this ? Authors should support this choice with some reference or results, even if in a supplementary form.

Line 143 “time averaged zonal” … for 2015 ? be explicit.

Comment on Figure 1, but general to all figures: Figures are with very small fonts. and axis must be corrected to °W/°E or °N/°S, not negative values. The vector density is to high and there is no "vector scale" in the figure. In general, a "pivot" colormap with zero white should be considered for velocity (negative blue, positive red). Use another colormap for salinity (suggestion: parula) and indicate the coastline in high resolution. Panels must be labeled (ex. A, B, C, D) and indicated in the legend.

Lines 166-167 ...“quite close to existing observations”. Authors should show this comparison with observations (for example, data from ARGOS floats), even if in a supplementary material.

Line 171 …”whole year”... is 2015, the reference year ? be explicit.

Comment on Figure 2 (and general figures): Same comments about the figure 1. Pivot color scale (zero white) and a better minor xticks. Also the isopycnal levels must be indicated "in the figure" as a text. The authors could explore a lot in this figure. Mainly the development of nSEC from November onwards. Discussions are missing in Figure 2, with several processes and important aspects about regional circulation that the authors could explore.

Comment on Figure 5: The use of the same color table for two different fields is not recommended. This hinders the identification of structures.

lines 217 - 222 - Here it would be interesting to show the temporal variability, the months and days chosen according to the PV and the extreme salinity, in complementary data that could be better understandable than the videos.

Lines 223-225 The authors argument is based on a snapshot. You could use a line plot to show evolution of avg PV and avg salinity (including variances) in selected boxes, for example.

Lines 276-277 …”(SEUC and NEUC/GUC) currents, at least up to 20 277oW.”. How long is the residence/persistence (time) of these structures ?

Comment on Figure 9: Same comment as the previous figures. Use different color scales for different fields in the same figure, and adjust the details of the maps. In this type of graph, the coastline should be highlighted and the latitudes and longitudes corrected on the map. Very hard to see what you are explaining in the figure.

Lines 337-339 Even though it is not part of the scope of the work, the authors should indicate that the tidal effect is insignificant (with refs).

Line 362 Section 3.4.3. Friction. Although NEMO has a known configuration, the authors little explore the effects of bottom topography, especially in the Gulf of Guinea. The explanations in section 3.4.3 do not seem convincing enough.

Comments on Figs 12-15: These figures are very difficult to see what the authors claim. The bathymetry is completely distorted by the composition of the map. Authors should reconsider the graphic production of the figures as it is only possible to "believe" the text.

Line 519-520 - This is the main goal of the study? “we have analyzed the ocean dynamics in the subsurface layer of the Gulf of Guinea”. Note that the manuscript changes its central line because it begins by speaking the Tropical Atlantic and in that part it goes entirely to the Gulf of Guinea. The suggestion is to make the central line of the article more defined. Both in the discussions and in the results the focus is always on the Gulf of Guinea, I think it would be interesting, in the introduction, or in the beginning of methodology (or even in the first part of the results), an initial map of the (study) area with the location of these geographic features, which would be more interesting to readers, just as the authors did in figure 1, in the vertical profile.

Round 2

Reviewer 2 Report

The authors adequately answered all the questions posed in the review and the manuscript has greatly improved. I agree with all the comments made and recommend the article for publication.

Thank you for the opportunity to review the manuscript.